# Prediction of Ship Trajectory in Nearby Port Waters Based on Attention Mechanism Model

Junhao Jiang [1] and Yi Zuo [1,2,*]

1    Navigation College, Dalian Maritime University, Dalian 116026, China
2    Maritime Big Data & Artificial Intelligent Application Centre, Dalian Maritime University, Dalian 116026, China
*    Correspondence: zuo@dlmu.edu.cn

**Abstract:** In recent years, the prediction of ship trajectory based on automatic identification system (AIS) data has become an important area of research. Among the existing studies, most focus on a single ship to extract features and train models for trajectory prediction. However, in a real situation, AIS contains a variety of ships and trajectories that need a general model to serve various cases. Therefore, in this paper, we include an attentional mechanism to train a multi-trajectory prediction model. There are three major processes in our model. Firstly, we improve the traditional density-based spatial clustering of applications with noise (DBSCAN) algorithm and apply it to trajectory clustering. According to the clustering process, ship trajectories can be automatically separated by groups. Secondly, we propose a feature extraction method based on a hierarchical clustering method for a trajectory group. According to the extraction process, typical trajectories can be obtained for individual groups. Thirdly, we propose a multi-trajectory prediction model based on an attentional mechanism. The proposed model was trained using typical trajectories and tested using original trajectories. In the experiments, we chose nearby port waters as the target, which contain various ships and trajectories, to validate our model. The experimental results show that the mean absolute errors (MAEs) of the model in longitude (°) and latitude (°) compared with the baseline methods were reduced by 8.69% and 6.12%.

**Keywords:** trajectory prediction; AIS data; feature extraction; attention mechanism; neural network

## 1. Introduction

Over the years, ships have become larger and more diversified, and the number of ships has also increased. This brings potential safety hazards to the navigation of ships [1]. According to a report on the causes of collisions, 95% are caused by humans [2]. It is insufficient for officers to rely solely on navigational information from the electronic chart display, information system (ECDIS), and automatic identification system (AIS) equipped on ships to assist in decision-making. The prediction of ship trajectory can provide correct auxiliary decision support for pilots. Therefore, accurate ship trajectory prediction is necessary for a ship navigation system to reduce the risk of ship accidents. Trajectory prediction is extensively studied in the field of ships. The models adopted can be divided into those using traditional methods and using machine learning. In traditional methods, the modeling of a ship's motion process is conducted through one or more sets of mathematical kinematic equations, taking all possible influencing factors (such as mass, force, inertia, yaw, and rate) into account and using physical laws to model and calculate the ship's motion characteristics. Best et al. proposed a ship curve model integrating linear motion, circular motion, and parabolic motion that can accurately describe a ship's motion characteristics [3]. Inoue et al. used a ship hydrodynamic model to predict ship trajectory, which can calculate ship movement according to actual ship data, such as hull, rudder, propulsion system, ocean current speed, and major details of wind [4]. In modeling that uses machine learning, it includes Kalman filtering (KF), support vector regression (SVR),

artificial neural network (ANN), etc. Liu et al. took the longitude and latitude, course, and speed from ship historical trajectory data as sample features and proposed a trajectory prediction model based on SVR [5]. Because an ANN benefits the learning of complex relationships, it can be used to learn complex space–time relationships among variables [6]. Gan et al. first used a K-means clustering algorithm to group ship historical trajectory and then used the grouping results to build an ANN model to predict ship trajectory [7].

In practical applications, trajectory navigation is a stochastics process via time, and most scholars pursue trajectory prediction using a time series analysis and approach. Therefore, many scholars have studied a series of prediction problems using nonlinear information, such as a long and short-term memory neural network (LSTM-NN) [8,9]. Ma et al. proposed a new trajectory feature representation method, which used a hierarchical clustering algorithm to analyze and extract ship navigation behaviors of multiple trajectories, and designed an integrated model for simultaneously predicting multiple trajectories based on LSTM-NN [10]. Gao et al. combined the advantages of a trajectory proposal network for motion prediction (TPNet) and LSTM-NN to realize the multistep prediction of ship trajectory, which is suitable for real-time analysis and has high accuracy [11]. The gated recurrent unit neural network (GRU-NN) is a variant of the LSTM-NN, which only requires updating and resetting of the gate to regulate the flow of information [12]. Han et al. proposed a short-term real-time trajectory coordinate point prediction method based on GRU-NN from the perspective of historical trajectory data and real-time trajectory data [13]. Then, a combined online learning model combining K-means clustering and GRU-NN was proposed for trajectory prediction [14]. In the model, a k-means algorithm is used to cluster the trajectory points adaptively, and the online learning prediction model based on GRU-NN is used to learn from the trajectory points of each cluster.

In practice, due to the randomness and diversity of disturbances, a ship's motion state changes frequently, making it difficult to find the change rule and feature extraction. The AIS data collection frequency is also different, resulting in the uneven distribution of the data time difference. The traditional single trajectory feature extraction method has high sensitivity in densely packed ship trajectory points, which makes it easy to lose local features and difficult to restore the original trajectory. Trajectory data are often nonlinear and occur over a certain space and time scale [15]. The traditional models mainly study the prediction of a single trajectory, and each trajectory requires a separate training model. Therefore, the lack of a long and accurate multi-trajectory sequence prediction model poses a challenge to current research.

To solve the above problems, a multi-trajectory feature extraction and prediction model is proposed. Firstly, we improved the traditional density-based spatial clustering of applications with noise (DBSCAN) algorithm and applied it to trajectory clustering. According to the clustering process, ship trajectories can be automatically separated into groups, thus generating different trajectory categories. In this work, we used the dynamic time warping (DTW) algorithm to measure the similarities among different trajectories. Secondly, we propose a feature extraction method based on a hierarchical clustering method for a trajectory group. This method improves the traditional single trajectory feature extraction method and proposes multi-trajectories using group hierarchical clustering (GHC) for feature extraction based on a trajectory group. According to the extraction process, typical trajectories can be obtained for individual groups. Thirdly, we propose a multi-trajectory prediction model based on an attentional mechanism. The proposed model was trained using typical trajectories and tested using original trajectories. In the experiments, we chose nearby port waters as the target, which contains various ships and trajectories, to validate our model.

In the experiments, MAE and RMSE were used to measure the performance of the proposed model, and we compared the features proposed by the GHC algorithm with those extracted by hierarchical clustering (HC) and Douglas–Peucker (DP) algorithms. The average similarity between our features and the original features was 2.267. Under the same conditions, the GHC algorithm was 3.74% better than the HC algorithm and

60.12% better than the DP algorithm. In the second part of the experiment, we studied the feature extraction based on GHC to train the trajectory prediction model. First, the experiment compared the advantages of the attention mechanism model in the feature prediction. The MAE of the longitude and latitude of the departure trajectory were 0.0198 and 0.0173, respectively, which were 8.65%, 16.26%, 15.48%, 19.58%, 6.82%,17.07%, 13.35% and 22.56% lower than the LSTM-NN, GRU-NN, bidirectional long and short term memory neural network (BiLSTM-NN) and bidirectional gate recurrent unit neural network (BiGRU-NN), respectively. The MAE of the longitude and latitude of the arrival trajectory were 0.00898 and 0.00673, respectively, which were 52.60%, 63.84%, 53.07%, 62.71%, 38.95%, 56.89%, 47.36% and 58.93% lower than LSTM-NN, GRU-NN, BiLSTM-NN and BiGRU-NN, respectively. This study also compared the effects of different feature extraction results on model training and found that the features extracted using the GHC algorithm were the best among different models.

The contributions of this article mainly include the following aspects. Firstly, we improved the traditional DBSCAN algorithm and applied it to trajectory clustering. According to the clustering process, the common trajectory features can be automatically extracted by group. Secondly, we proposed a feature extraction method based on the hierarchical clustering method for a trajectory group. According to the extraction process, typical trajectory features can be obtained for individual groups. Thirdly, we proposed a multi-trajectory prediction model based on an attentional mechanism. Compared to the traditional prediction models, it has a higher accuracy and better generalization. In addition, the high accuracy of trajectory prediction is not only of benefit for navigation path planning, and also greatly contributes to ship collision avoidance and route optimization so as to enhance navigation safety and improve management efficiency for the sustainable shipping development.

The remainder of this article is organized as follows. Section 2 reviews studies on traditional feature extraction methods and trajectory prediction models. Section 3 provides the methodology for the introduction of the DBSCAN clustering method, the improved hierarchical clustering feature extraction method, and the multi-trajectory prediction model based on an attention mechanism. Section 4 describes the data and verifies the performance of the proposed model. Section 5 presents the conclusion of this paper and possible research in the future.

## 2. Background and Related Studies

### 2.1. Related Studies of the Trajectory Feature Extraction

The DP algorithm is a classic approach for the thinning of linear elements, which can handle a large number of aggregate data points in order to simplify the data volume [16]. At present, the DP algorithm has extensive application in ship trajectory compression and feature extraction. The core idea of the DP algorithm is to connect the starting and ending points of the curve and then determine the maximum vertical distance of the middle point to the connecting line, compared with a threshold value that has been set [17]. If it is larger than the threshold, it is divided into two segments and calculated again; if it is smaller than the threshold, the intermediate point is deleted to complete the feature extraction. See Appendix A and Figure A1 for detailed steps of the calculation and charts.

The hierarchical clustering algorithm is also an effective way to extract ship trajectory features. It first determines the number, $k$, of feature extraction points and calculates the Euclidean metric. The average value of the two points from the minimum Euclidean metric is taken to merge [18]. See Appendix A and Figure A2 for detailed steps of the calculation and charts. In addition, some studies have utilized other clustering methods for feature extraction, such as DBSCAN. However, there is currently a lack of research on feature extraction algorithms for trajectory groups.

### 2.2. Related Studies of the Trajectory Prediction Method

In this section, we review several related studies on the ship trajectory prediction method. A ship's driving operation is based on trajectory information within a period of time, rather than instantaneous information at a certain time step. In the field of machine learning, this information that lasts for a period of time can be regarded as time series information. Neural networks are widely used in processing time series data.

Recurrent neural network (RNN) is a network used to capture information from time series. The most significant feature of an RNN is that weight connections are also established among neurons within the layer [19]. Due to the inherent structure of an RNN, the RNN model has problems such as vanishing gradients, exploding gradients, sensitivity to noise, and a requirement of a large amount of training data. Therefore, an LSTM neural network was proposed by adding a gate mechanism on the basis of RNN, and this became a popular method in trajectory prediction [10,11,20]. An LSTM is a type of neural network that captures dynamic information in serial data [21,22]. It solves the problems of gradient disappearance and the long-term dependence of traditional neural networks by constructing specialized memory storage units and designing time back propagation methods [23–25]. Because of the complex structure inside an LSTM, the training time of an LSTM is long. A GRU is a new variant of short-term memory networks and long-term memory networks. It solves the problems existing in common RNNs and has good adaptability to the processing of a large number of nonlinear implementation sequences. Compared with an LSTM, the GRU has a simpler structure, fewer built-in parameters, and a faster training time [26]. Some scholars have also applied GRU to trajectory prediction, as shown in the literature [27]. The principles and pictures of the above methods are shown in Appendix A and Figures A3–A5. In addition, in order to consider the information of the forward and backward directions, bidirectional neural network mechanisms have emerged, such as BiLSTM and BiGRU. However, the prediction of multiple trajectories is lacking in the above literature.

### 3. Methodology

In order to extract feature points from trajectory data, better restore historical trajectories, and improve the accuracy of multi-trajectory prediction, a multi-trajectory feature extraction and prediction model is proposed. The process of the model is shown in Figure 1. As described, the model mainly consists of three parts. First, the data is processed by the data processing module and divided into trajectories. Second, feature extraction is performed for the trajectory. Third, the extracted features are used to train the machine learning model.

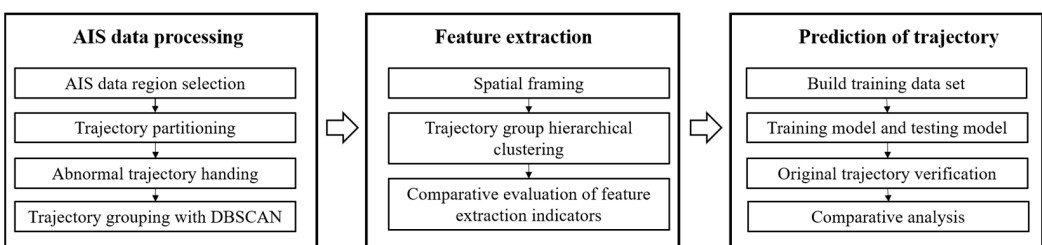

**Figure 1.** Overview of the multi-trajectory feature extraction and prediction framework.

### 3.1. Application of DBSCAN in Trajectory Clustering

In a given sea area, the ship trajectory has different types of motion modes and trajectory lengths. When the trajectory of the selected sea area is input directly into the model, it is difficult to obtain high-precision prediction results. In this study, the ship trajectories are grouped into clusters to identify ship motion patterns. Through this process, the ship trajectories within a group of species become more similar.

The DBSACN algorithm is a density-based clustering algorithm that can find arbitrarily shaped clusters in a noisy environment and automatically determine the number of clusters [28,29]. The traditional DBSCAN algorithm is shown in Figure 2a. In practical applications, DBSCAN is mainly applied to point clustering and cannot be used for the direct clustering analysis of ship trajectories. Ship trajectory is a set of discrete points, and the lines between the front and back points constitute trajectory segments. The DTW can be used to measure the similarity between trajectories of different lengths [30]. Therefore, in this work, the DBSCAN used DTW as the similarity measure to extend the traditional point-clustering analogy to widely linear clustering, thus extending it to trajectory clustering, as shown in Figure 2b. The main ideas are as follows.

(1) Define the neighborhood.
Select any trajectory $l_i$ in the trajectory set and define the neighborhood of trajectory $l_i$ as:

$$N_\varepsilon(l_i) = \{l_i \epsilon L | Dis(l_i, l_j) \leq \varepsilon\} \tag{1}$$

where $L$ is the trajectory set in the study area, and the neighborhood set of trajectory $l_i$ consists of trajectory not more than $\varepsilon$ in similarity.

(2) Determine the trajectory types.

    (A) Core trajectory: for the trajectory $l_i \epsilon L$, if $l_i$ is full $Num(N_\varepsilon(l_i)) \geq Num_{min}$, then the trajectory $l_i$ is the core trajectory. In Equation (1), $Num(N_\varepsilon(l_i))$ is the number of trajectories, wherein trajectory $l_i$ is doing so in the neighborhood, and $Num_{min}$ is the density threshold when trajectory $l_i$ is doing so in the neighborhood;

    (B) Boundary trajectory: if $l_i$ is full $Num(N_\varepsilon(l_i)) < Num_{min}$ and the trajectory is doing so within a similar threshold at a core trajectory, it is a boundary trajectory;

    (C) Noise trajectory: if $l_i$ is full $Num(N_\varepsilon(l_i)) < Num_{min}$ and the trajectory is not doing so under the threshold of any core trajectory, it is a noise trajectory.

(3) Traversing the trajectory set $N_\varepsilon(l_i)$ in the neighborhood of trajectory $l_i$: If they are not assigned to a cluster, assign the cluster label of trajectory $l_i$ to them. If they are core trajectories, access their neighborhood trajectories in turn until there are no more core trajectories in the neighborhood. Repeat the above steps until all trajectories have cluster labels.

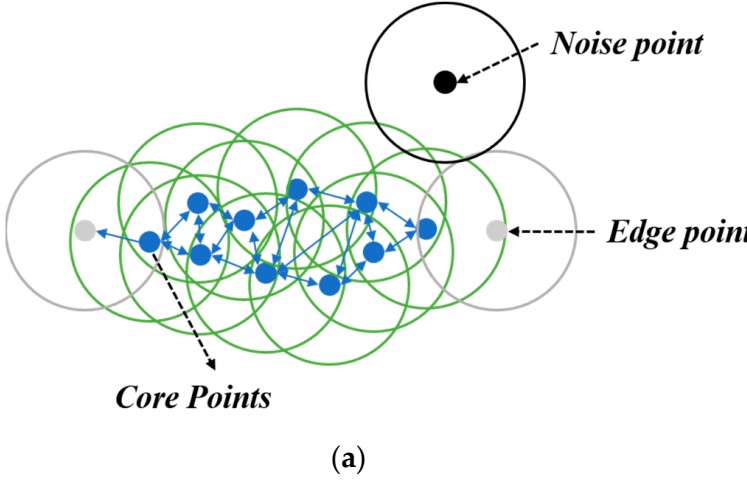

(a)

**Figure 2.** *Cont.*

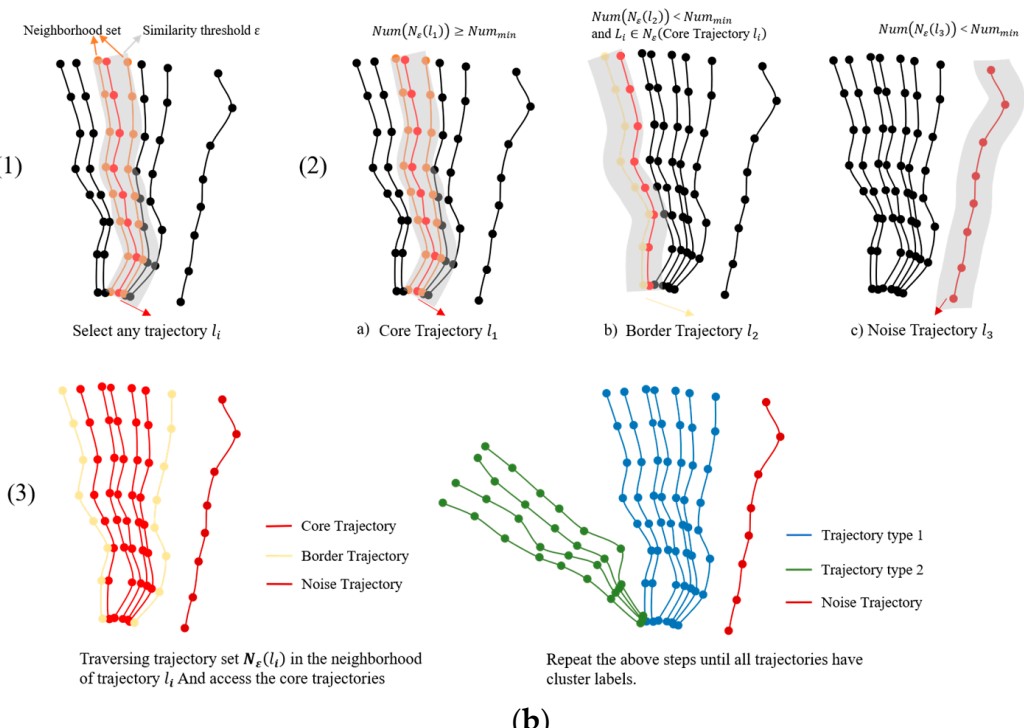

**Figure 2.** DBSCAN schematic diagram: (**a**) original DBSCAN; (**b**) DBSCAN applied to the trajectory.

*3.2. Feature Extraction*

3.2.1. Framing in Trajectory

Due to the different acquisition frequencies and time points of the AIS system, the data points contained in the trajectories of the same length are different. In this study, the idea of frame segmentation is adopted to ensure the unity of the points of the subsequent multi-trajectory feature extraction data. The trajectories in the same group are similar. Because of the difference in the sailing speeds and collection times, the frame segmentation in the time domain cannot guarantee the unity of the trajectory points. Therefore, this study adopted the method of spatial domain framing. As shown in Figure 3, the frame length is marked. Since the number of data points in a frame is not constant, the average value of the data in each frame is studied to represent the size of the data in the frame.

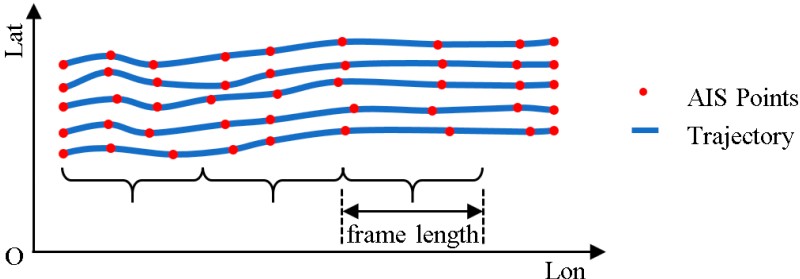

**Figure 3.** The paradigm of the trajectory AIS points for framing diagram.

3.2.2. Hierarchical Clustering Based on the Trajectory Groups

In order to solve the defect that the HC algorithm is sensitive to the turning density point and easily distort, in this paper, the trajectory group based on the hierarchical clustering algorithm was used to improve the traditional HC algorithm. The similarity is measured using Euclidean metric, As shown in Equation (2).

$$d_i = dis(p_i, p_{i+1}) = \sqrt{(x_i - x_{i+1})^2 + (y_i - y_{i+1})^2} \qquad (2)$$

where $d_i$ represents the Euclidean metric between $p_i$ and $p_{i+1}$.

A diagram is shown in Figure 4, and the specific steps are as follows:

(a) Calculate the similarity $d_i$ between adjacent two frames of each trajectory.
(b) Calculate the average similarity $AVG(d_i)$ between two adjacent frames in the same frame area and save it in the similarity set $S$. The specific is shown in Equation (3).

$$AVG(d_i) = \frac{\sum_{i=1}^{n-1} d_i}{n} \qquad (3)$$

where $n$ is the total number of trajectory points.

(c) The two points with the minimum similarity in the set $S$ are averaged and merged into the same point, replacing the original point in the trajectory set;
(d) Repeat steps a–c until the number of trajectory points converges to the target point $k$.

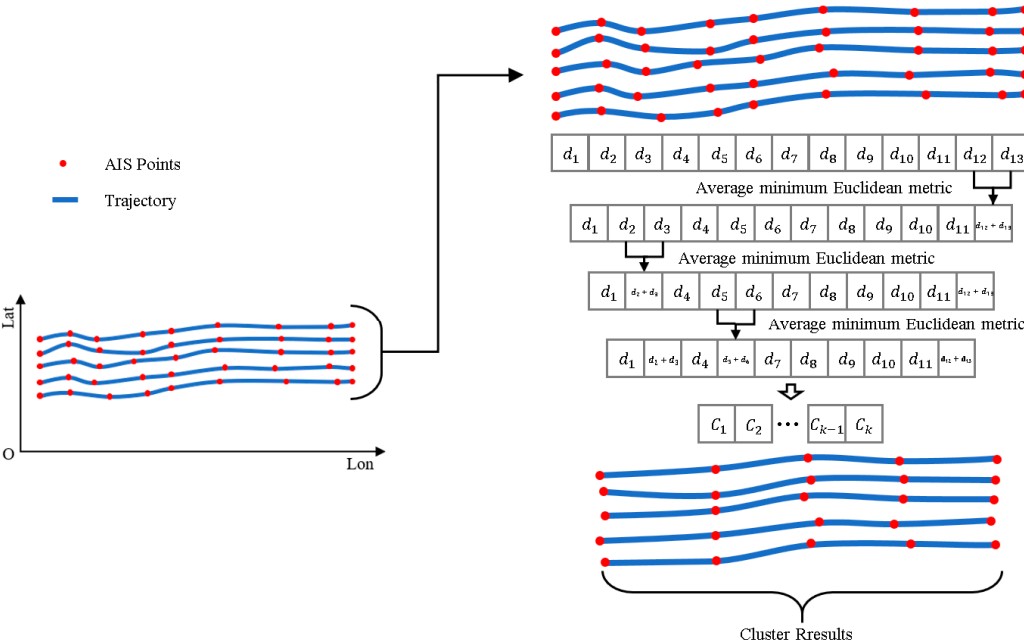

**Figure 4.** The schematic paradigm of multi-trajectory feature extraction using Euclidean metric.

The GHC algorithm considers the similarity of the trajectories within a grouping and uses the average similarity of all trajectories between two adjacent trajectory points (i.e., the similarity between the different frames) as the criterion for the trajectory point combination. The advantage of this method is that the trajectory feature extraction is no longer affected by the dense area of the trajectory points during the extraction of a single trajectory feature. It comprehensively measures the dense area of the trajectory within the trajectory group and better preserves the trajectory points that turn to the dense area, and the trajectory restoration degree is higher.

### 3.3. Multi-Trajectory Prediction Based on an Attention Mechanism

The attention mechanism in the deep learning model originates from the human brain mechanism [31]. It is widely used in computer vision and natural language processing tasks [32–34]. The attention mechanism assigns attention to input weights that address the characteristics of the target we want to detect. The accuracy of the model can be improved by using the attention mechanism. In addition, it is more efficient than traditional neural networks (such as an RNN and LSTM) in processing large-scale data [8,9]. Its core idea is to

allocate different weights to the state of the hidden layer by reasonably allocating attention to different input information to highlight the influence of important information on the results [35]. Attention mechanics are widely used in transformer (TRM) models. Part of the structure in the transformer was adopted in this study. The expression of the attention mechanism is shown in Equations (4) and (5).

$$f(x_i, y) = (W_1 * x_i, W_2 * y) \tag{4}$$

$$Attention = \sum_{i=1}^{n} softmax(f(x_i, y)) * x_i \tag{5}$$

where $x_i$ represents the input trajectory sequence series data, which is mapped to the interval $(0, 1)$ through the normalized exponential function (Figure 5).

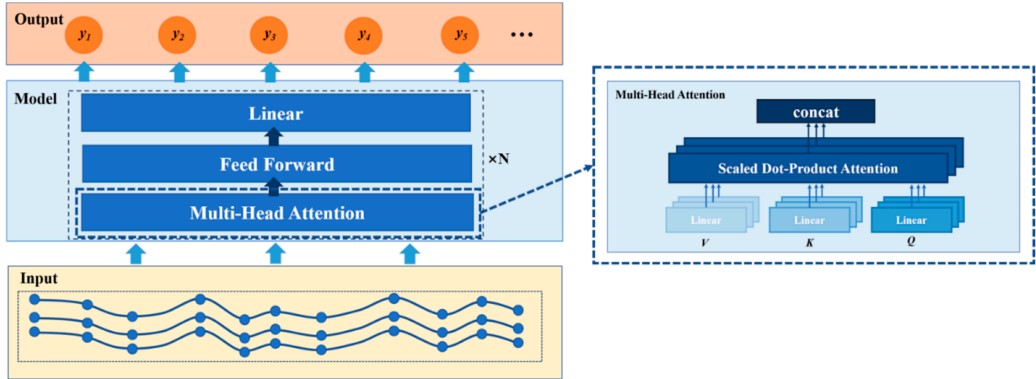

**Figure 5.** Ship trajectory prediction model based on the attention mechanism.

The multihead attention mechanism is a combination of multiple self-attention structures. Using key, query, and value, the multihead attention mechanism calculates the weight coefficient of the correlation and then weights the sum. The multihead attention mechanism is repeated many times by transforming the phenomena of key, query, and value and inserting them into the zoom point to receive attention. The linear change parameters $W$ of $Q$, $K$, and $V$ of each iteration are independent of each other and are not shared with each other. The specific calculation using Equations (6) and (7) are shown as follows:

$$Attention(Q, K, V) = softmax\left(\frac{QK^T}{\sqrt{d_k}}\right)V \tag{6}$$

$$MultiHead(Q, K, V) = (head_1 \oplus head_2 \oplus head_3 \oplus \cdots \oplus head_h) \tag{7}$$

where $Q$, $K$, and $V$ are the query vector, key vector, and value vector, respectively; $d_k$ is the dimension of $K$; and $h$ represents the number of heads of multiple attention mechanisms. From Equations (6) and (7), we can observe that the multihead attention mechanism can be seen as a combination of multiple attention models and as a trajectory information weight distribution scheme, which can allow the model to more fully extract trajectory information during prediction. Finally, the result of the calculation of the output value attention layer is input into the feedforward neural network, and the predicted value of the trajectory point is obtained through linear transformation.

The ship's navigation information comes from AIS data, including longitude, latitude, speed over ground, and course over ground. Therefore, the navigation information of ship $i$ at time $j$ can be described as:

$$P_{ij} = \left[Lon_{i,j}, Lat_{i,j}, SOG_{i,j}, COG_{i,j}\right] \tag{8}$$

The dynamic navigation information of the ship within a certain period of time can be expressed as $p_{i,j-n+1}, p_{i,j-n+2}, \ldots, p_{i,j}$, and $p_{i,j+1}$. The input data in the model are shown in Equation (9):

$$P = [[P_1], [P_2], [P_3], \ldots, [P_n]] = \left[\begin{bmatrix} p_{1,1} \\ p_{1,2} \\ \vdots \\ p_{1,k} \end{bmatrix}, \begin{bmatrix} p_{2,1} \\ p_{2,2} \\ \vdots \\ p_{2,k} \end{bmatrix}, \begin{bmatrix} p_{3,1} \\ p_{3,2} \\ \vdots \\ p_{3,k} \end{bmatrix}, \ldots, \begin{bmatrix} p_{n,1} \\ p_{n,2} \\ \vdots \\ p_{n,k} \end{bmatrix}\right] \tag{9}$$

where $n$ represents the number of input trajectory sets, and $k$ represents the number of trajectory points in trajectory sets.

The ship multi-trajectory prediction model can be defined as:

$$Y_j' = F([P_1], [P_2], [P_3], \ldots, [P_n]) \tag{10}$$

In this paper, the parameters of the model are directly related to the input trajectory sequence data, and the parameters need to be adjusted to better adapt to this situation.

## 4. Experiment

### 4.1. Data Description

In order to test the algorithm's performance in the proposed multi-trajectory feature extraction and prediction framework, in this study, the AIS data collected from waters near Yantai Port in Bohai Region of China from January to June 2019 were selected as analysis objects, among which the longitude range was [121.00°, 122.00°] and latitude range was [37.58°, 38.00°]. The reason is that the harbor waters have a variety of trajectory navigation channels, which can be divided into different types of trajectories in the same geographical area. These trajectories are very suitable for testing the overall performance of the proposed algorithm. The selected trajectory is shown in Figure 6. A total of 334 arrival trajectories and 349 departure trajectories were screened. As can be seen from the processed trajectory diagram, different trajectories in the same area had similarities, although there were some differences in the details, and the trajectories had an obvious grouping phenomenon.

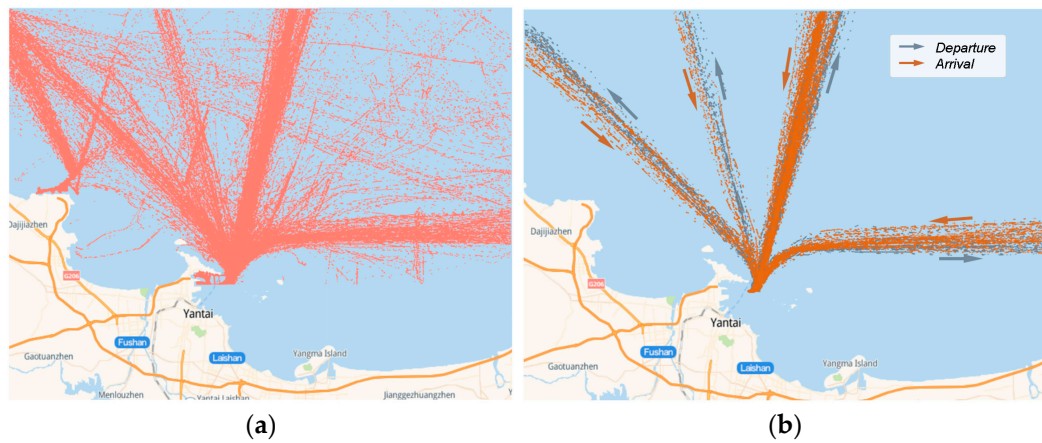

**Figure 6.** AIS data map of the selected sea area: (**a**) original data; (**b**) data after data processing.

### 4.2. Evaluation Indicators

R-squared, mean absolute error (MAE), and root mean square error (RMSE) are commonly used evaluation indexes [36]. R-squared is an indicator for the evaluation of the degree of fit. MAE is the mean of the absolute errors [37]. RMSE is the square root of the mean square difference between the predicted value and the actual observation [38]. MAE and RMSE are used to measure the difference between predicted and actual values [39]. R-squared, MAE, and RMSE are calculated according to Equations (11)–(13),

$$R^2 = 1 - \frac{\sum_{i=1}^{n}\left(y_i - y_i'\right)^2}{\sum_{i=1}^{n}\left(\overline{y} - y_i'\right)^2} \tag{11}$$

$$MAE = \frac{1}{N}\sum_{i=1}^{n}|y_i - y_i'| \tag{12}$$

$$RMSE = \sqrt{\frac{1}{N}\sum_{i=1}^{n}(y_i - y_i')^2} \tag{13}$$

where $N$ is the sample quantity, $\overline{y}$ is sample average, $y_i'$ is the forecast trajectory points, and $y_i$ is the real value of the trajectory. The closer $R^2$ is to 1, the better the model fits. The smaller the *MAE* and *RMSE* values, the closer the predicted values are to the real values, and the higher the prediction accuracy of the model.

### 4.3. The Result of the DBSCAN and Data Analysis

In this study, the DTW algorithm was used as the similarity measurement method of DBSCAN, and similar ship trajectories were clustered. The clustering results of the departure trajectory and arrival trajectory are shown in Figure 7. According to the clustering results, the trajectory of the port departure and arrival can be roughly divided into four trajectory types, with obvious differences among the different trajectory groups and roughly the same shape among the same trajectory groups. Table 1 shows the clustering results of the trajectories, including labels, numbers, and clustering proportions of each cluster. It can be seen from Table 1 that the cluster of type 3 contained the largest number of trajectories in the departure and arrival trajectories, accounting for 54.19% and 55.01%, respectively. The number of trajectories of type 2 was the least, accounting for 5.69% and 5.44%, respectively. The noise trajectories accounted for 2.96% and 2.29%, respectively.

**Table 1.** Clustering results of the trajectories.

| Direction of Movement | Trajectory Type | Quantity | Ratio (%) |
|---|---|---|---|
| Departure | Type D1 | 87 | 24.93 |
| | Type D2 | 19 | 5.44 |
| | Type D3 | 192 | 55.01 |
| | Type D4 | 43 | 12.32 |
| | Noise | 8 | 2.29 |
| Arrival | Type A1 | 85 | 25.45 |
| | Type A2 | 19 | 5.69 |
| | Type A3 | 181 | 54.19 |
| | Type A4 | 40 | 11.98 |
| | Noise | 9 | 2.69 |

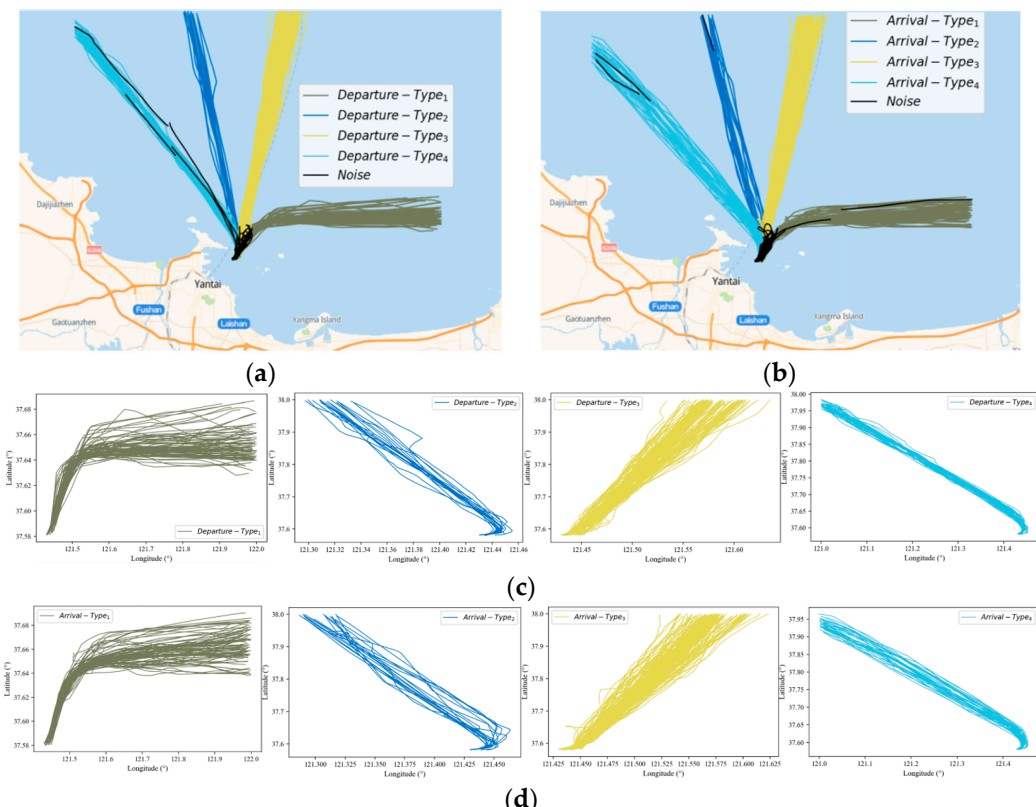

**Figure 7.** Clustering results of the departure and arrival trajectories: (**a**,**b**) visualize the clustering results in map; (**c**,**d**) visualize the clustering results in longitude and latitude axes.

In order to verify the superiority of the DBSCAN's performance in track grouping applications, we compared it with the K-medoids algorithm. The clustering effect of the departure and arrival trajectory is shown in Figure 8. It can be clearly seen from the figure that the clustering effect of the K-medoids algorithm is poor. One reason for this is that when the K-clustering centers are randomly selected, it is difficult to separate the noise trajectories within the trajectories, which has an impact on the selection process of the clustering centers. The DBSCAN algorithm can effectively eliminate the noise trajectory, as shown in Figure 8a,b. In addition, in order to quantify the clustering effect, the silhouette coefficient was adopted as the evaluation index, which is specifically described in Equation (14):

$$s(i) = \frac{b(i) - a(i)}{\max\{a(i), b(i)\}} \tag{14}$$

where $a(i)$ represents the average similarity between vector $i$ and other targets in the same cluster; $b(i)$ is the average similarity between vector $i$ and all points in a cluster in which it is not contained. The value range of the silhouette coefficient is −1 to 1, and the closer it is to 1, the cohesiveness and separation degree are relatively superior [40]. The silhouette coefficient table is shown in Table 2. It can be seen from the table that the DBSCAN algorithm had better evaluation indexes than K-medoids. It can be seen from the parameters that the algorithm has obvious advantages in the application of trajectory grouping.

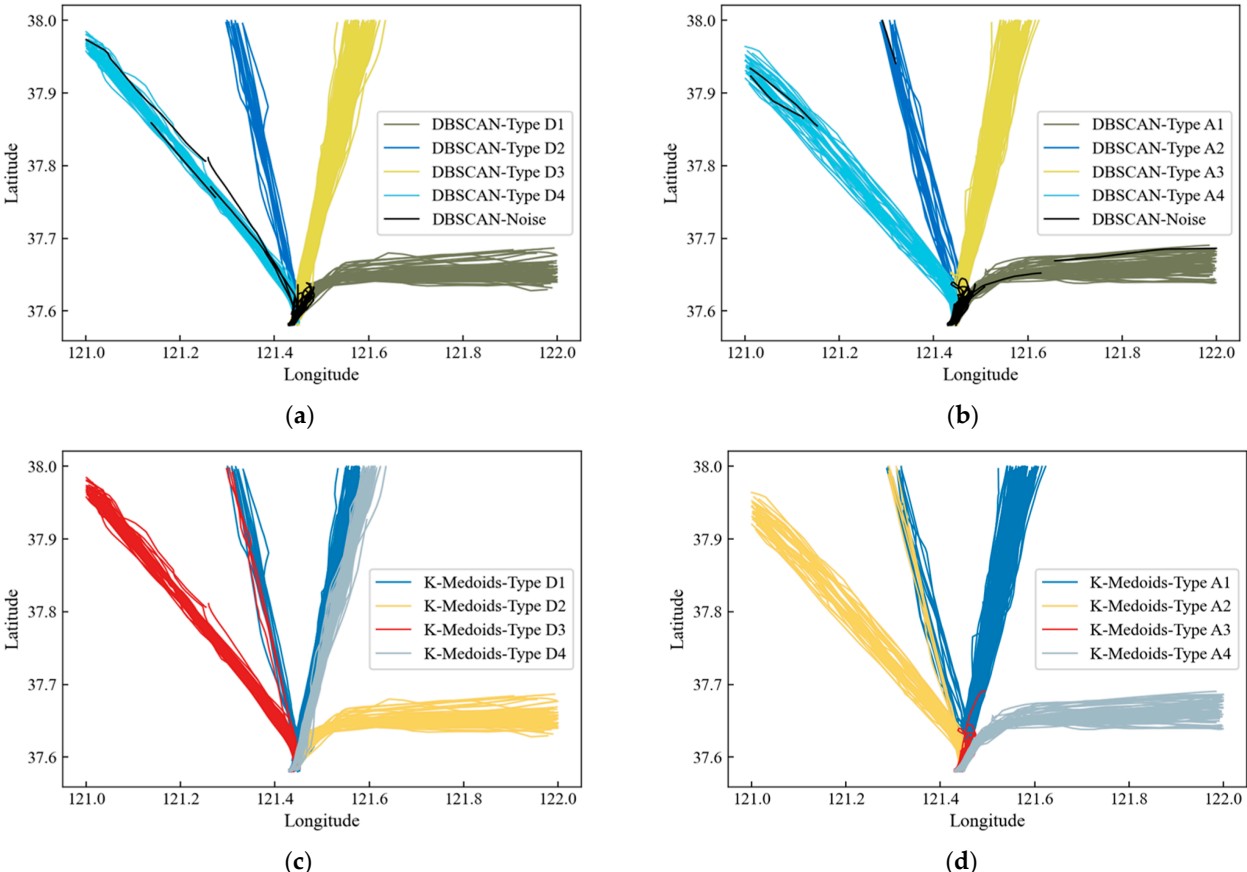

**Figure 8.** Comparison of the clustering algorithm effects: (**a**) DBSCAN for the departure trajectory; (**b**) DBSCAN for the arrival trajectory; (**c**) K-medoids for the departure trajectory; (**d**) K-medoids for the arrival trajectory.

**Table 2.** Silhouette coefficient of the different clustering methods.

| Direction of Movement | Silhouette Coefficient | |
|---|---|---|
| | **DBSCAN** | **K-Medoids** |
| Departure | 0.881 | −0.0417 |
| Arrival | 0.872 | −0.0667 |

The distribution of the course and speed of four different types of trajectory departures and arrivals is shown in Figure 9. Figure 9a shows the distribution diagram of various characteristics of the departure trajectory. Figure 9b shows the distribution diagram of various characteristics of the arrival trajectory. It can be seen from the clustering results and feature result graphs that the departure and arrival trajectory type 1 was the steering trajectory, and the course and speed had a certain variation trend. In departure trajectory type 1, the course was mainly between 260° and 280°, and speeds of 7 to 12 knots and 17 to 20 knots were more common. In arrival trajectory type 1, the course was mainly distributed between 80° and 100°, and the speed was mainly 9 to 13 knots and 18 to 21 knots. The trajectory types corresponding to other departure and arrival trajectories were straighter, and the course distribution was more concentrated. In most cases, the speed was also relatively concentrated. In the departure trajectory type 2, the course distribution was between 150° and 170°, and the speed distribution was between 9 and 13 knots. In arrival trajectory type 2, the course distribution was between 340° and 350°, and the speed was mainly between 7 and 12 knots. In departure trajectory type 3, the course distribution was between 180° and 200°, and the speed distribution was between 7.5 and 10 knots

and between 13 and 17 knots. In the approach trajectory type 3, the course distribution was between 0° and 30°, and the speed was mainly between 8 and 10.5 knots and 13 and 17 knots. In departure trajectory type 4, the course was mainly distributed between 125° and 140°, and the speed was mainly distributed between 7.5 and 12 knots. In arrival trajectory type 4, the course was mainly distributed between 305° and 330°, and the speed was mainly distributed between 7 and 12 knots, with relatively large speed variations.

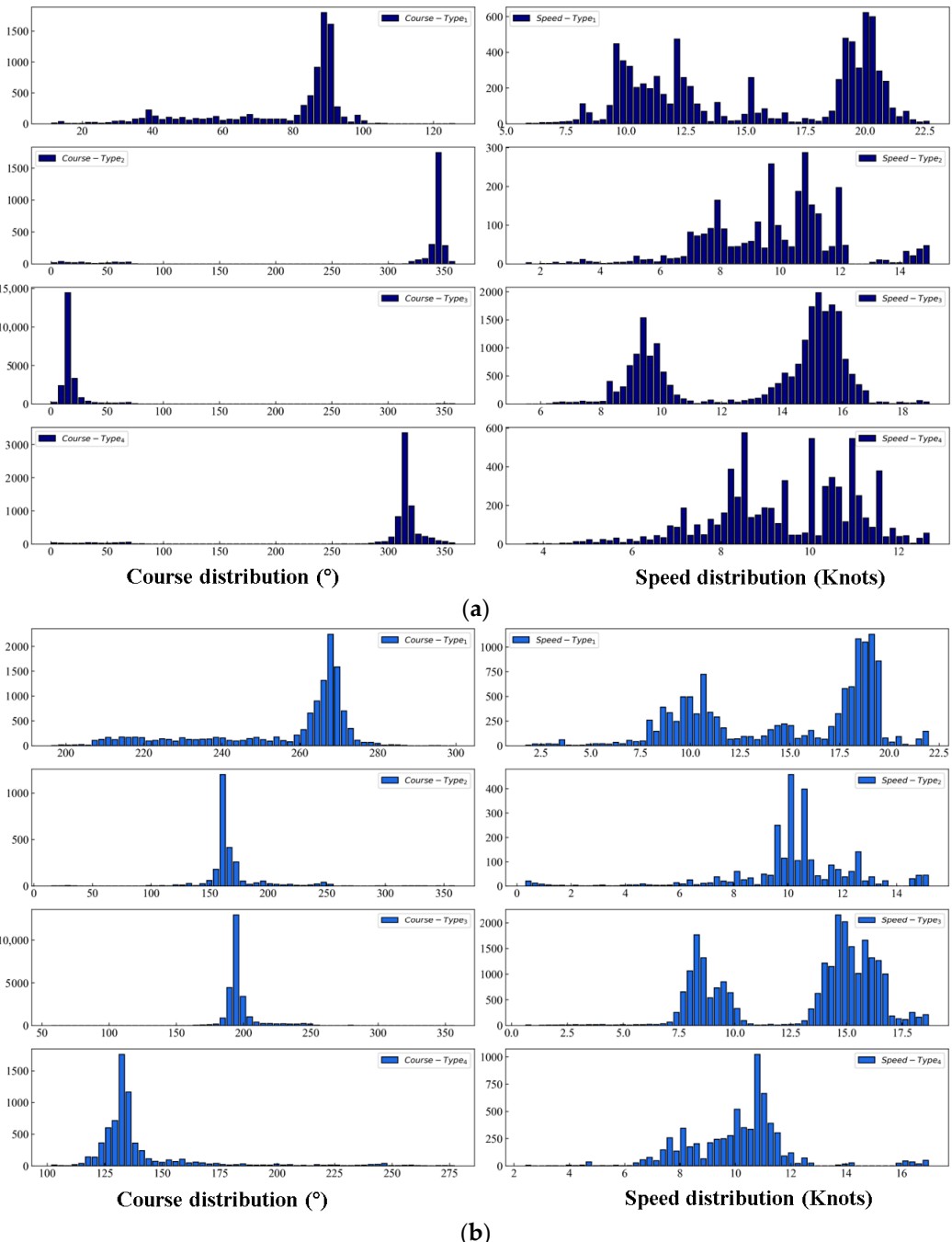

**Figure 9.** Distribution of the arrival and departure trajectory data: (**a**) departure data distribution; (**b**) arrival data distribution.

*4.4. The Result of the Feature Extraction*

4.4.1. The Result of the Framing

In order to reduce the sensitivity of the trajectory density points and the input data for the subsequent model construction, all trajectory lengths should be guaranteed to be the

same. Therefore, this study adopted the method of frame segmentation to process the data. The time difference before and after the trajectory points is shown in Figure 10. As can be seen from the figure, the time difference between the departure and arrival trajectories was mostly concentrated between 0 and 20 s, and a small amount of data was distributed between 200 and 300 s, indicating that there was a certain uneven distribution of the data collection times.

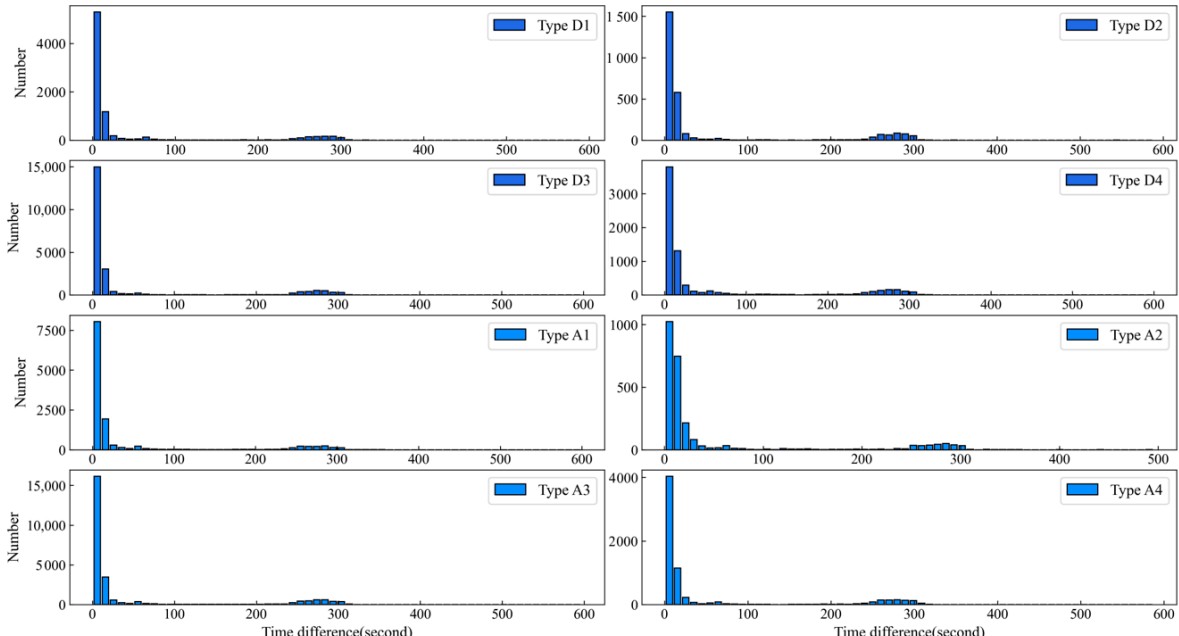

**Figure 10.** Distribution of the trajectory points over time.

As can be seen from above, frame splitting can ensure that each trajectory had the same trajectory length. In this study, the frame length was set as 0.005° to 0.010°, and the interval was 0.005°. The variance of the number of trajectory points of each trajectory contained by the trajectory group with different frame lengths is shown in Figure 11. We selected the frame length whose variance was 0 for the first time as the frame dividing length. The results of the variance of the different frames are shown in Table 3.The optimal frame length and the number of trajectory points after the framing of each trajectory group are shown in Table 4.

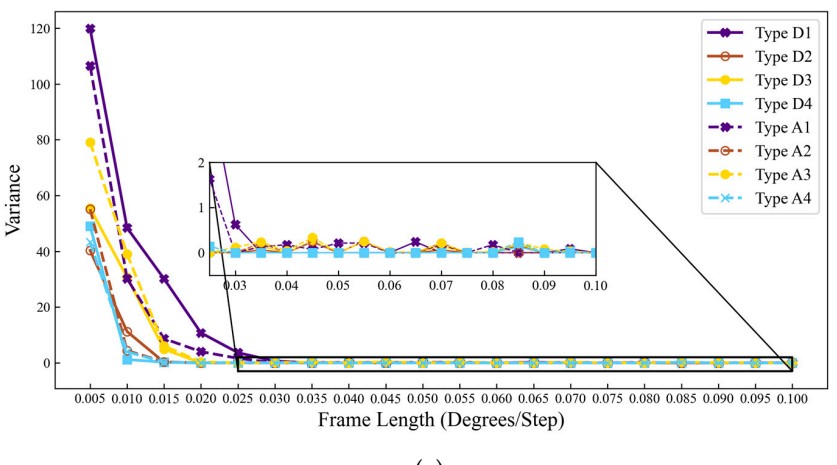

(**a**)

**Figure 11.** *Cont.*

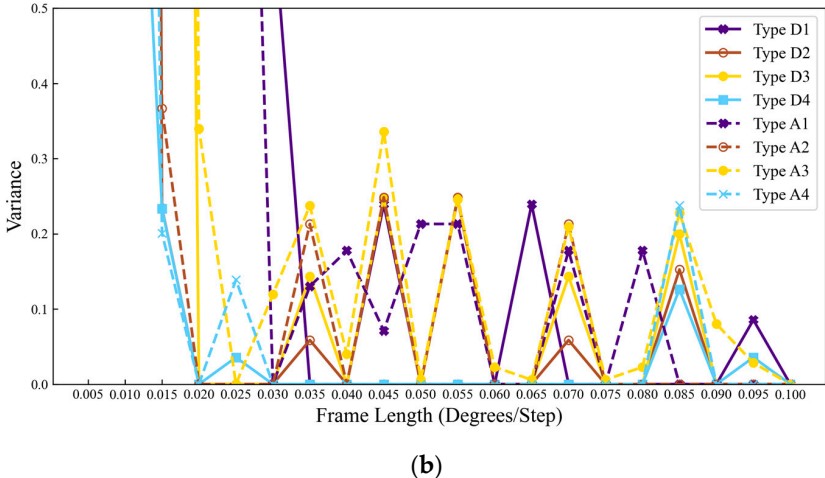

(**b**)

**Figure 11.** Variance of the number of trajectory points of each trajectory included in the trajectory group under different frame lengths: (**a**) Comparison results of frame length from 0.005 to 0.100 with 0.005 interval; (**b**) Detailed results with the variance from 0.0 to 0.5 with 0.1 interval.

**Table 3.** Variance results of the different frames.

| Degree | Departure | | | | Arrival | | | |
|---|---|---|---|---|---|---|---|---|
| | Type D1 | Type D2 | Type D3 | Type D4 | Type A1 | Type A2 | Type A3 | Type A4 |
| 0.005 | 119.83 | 40.31 | 55.47 | 48.98 | 106.56 | 55.10 | 79.14 | 43.35 |
| 0.010 | 48.41 | 11.23 | 30.67 | 1.21 | 30.22 | 4.40 | 39.03 | 3.99 |
| 0.015 | 30.15 | 0.23 | 4.90 | 0.23 | 8.67 | 0.37 | 6.06 | 0.20 |
| 0.020 | 10.75 | 0.00 | 0.00 | 0.00 | 4.02 | 0.00 | 0.34 | 0.00 |
| 0.025 | 3.66 | 0.00 | 0.00 | 0.04 | 1.63 | 0.00 | 0.00 | 0.14 |
| 0.030 | 0.63 | 0.00 | 0.00 | 0.00 | 0.00 | 0.00 | 0.12 | 0.00 |
| 0.035 | 0.00 | 0.06 | 0.14 | 0.00 | 0.13 | 0.21 | 0.24 | 0.00 |
| 0.040 | 0.00 | 0.00 | 0.00 | 0.00 | 0.18 | 0.00 | 0.04 | 0.00 |
| 0.045 | 0.24 | 0.25 | 0.25 | 0.00 | 0.07 | 0.25 | 0.34 | 0.00 |
| 0.050 | 0.00 | 0.00 | 0.00 | 0.00 | 0.21 | 0.00 | 0.01 | 0.00 |
| 0.055 | 0.00 | 0.25 | 0.25 | 0.00 | 0.21 | 0.25 | 0.25 | 0.00 |
| 0.060 | 0.00 | 0.00 | 0.00 | 0.00 | 0.00 | 0.00 | 0.02 | 0.00 |
| 0.065 | 0.24 | 0.00 | 0.00 | 0.00 | 0.00 | 0.00 | 0.01 | 0.00 |
| 0.070 | 0.00 | 0.06 | 0.14 | 0.00 | 0.18 | 0.21 | 0.21 | 0.00 |
| 0.075 | 0.00 | 0.00 | 0.00 | 0.00 | 0.00 | 0.00 | 0.01 | 0.00 |
| 0.080 | 0.00 | 0.00 | 0.00 | 0.00 | 0.18 | 0.00 | 0.02 | 0.00 |
| 0.085 | 0.00 | 0.15 | 0.20 | 0.13 | 0.00 | 0.00 | 0.23 | 0.24 |
| 0.090 | 0.00 | 0.00 | 0.00 | 0.00 | 0.00 | 0.00 | 0.08 | 0.00 |
| 0.095 | 0.09 | 0.00 | 0.00 | 0.04 | 0.00 | 0.00 | 0.03 | 0.00 |
| 0.100 | 0.00 | 0.00 | 0.00 | 0.00 | 0.00 | 0.00 | 0.00 | 0.00 |

**Table 4.** Optimal frame length and the number of trajectory points after the framing of each group of trajectories.

| Direction of Movement | Trajectory Type | Frame Length (Degree) | Trajectory Number |
|---|---|---|---|
| Departure | Type D1 | 0.035 | 17 |
| | Type D2 | 0.020 | 21 |
| | Type D3 | 0.020 | 21 |
| | Type D4 | 0.020 | 20 |
| Arrival | Type A1 | 0.030 | 19 |
| | Type A2 | 0.020 | 20 |
| | Type A3 | 0.025 | 17 |
| | Type A4 | 0.020 | 18 |

### 4.4.2. Analysis of the Feature Extraction Results

This section discusses the performance of the different feature extraction methods. According to the above, the DTW algorithm can be used as an evaluation index to measure the similarity of the time series features. The smaller the value, the greater the similarity them. Figure 12 shows a comparison between the departure and arrival trajectories using different feature extraction methods and the original trajectories. The figure includes locally enlarged images of various feature extraction methods. Figure 13 shows a similarity comparison diagram of the feature extraction of the departure and arrival trajectories.

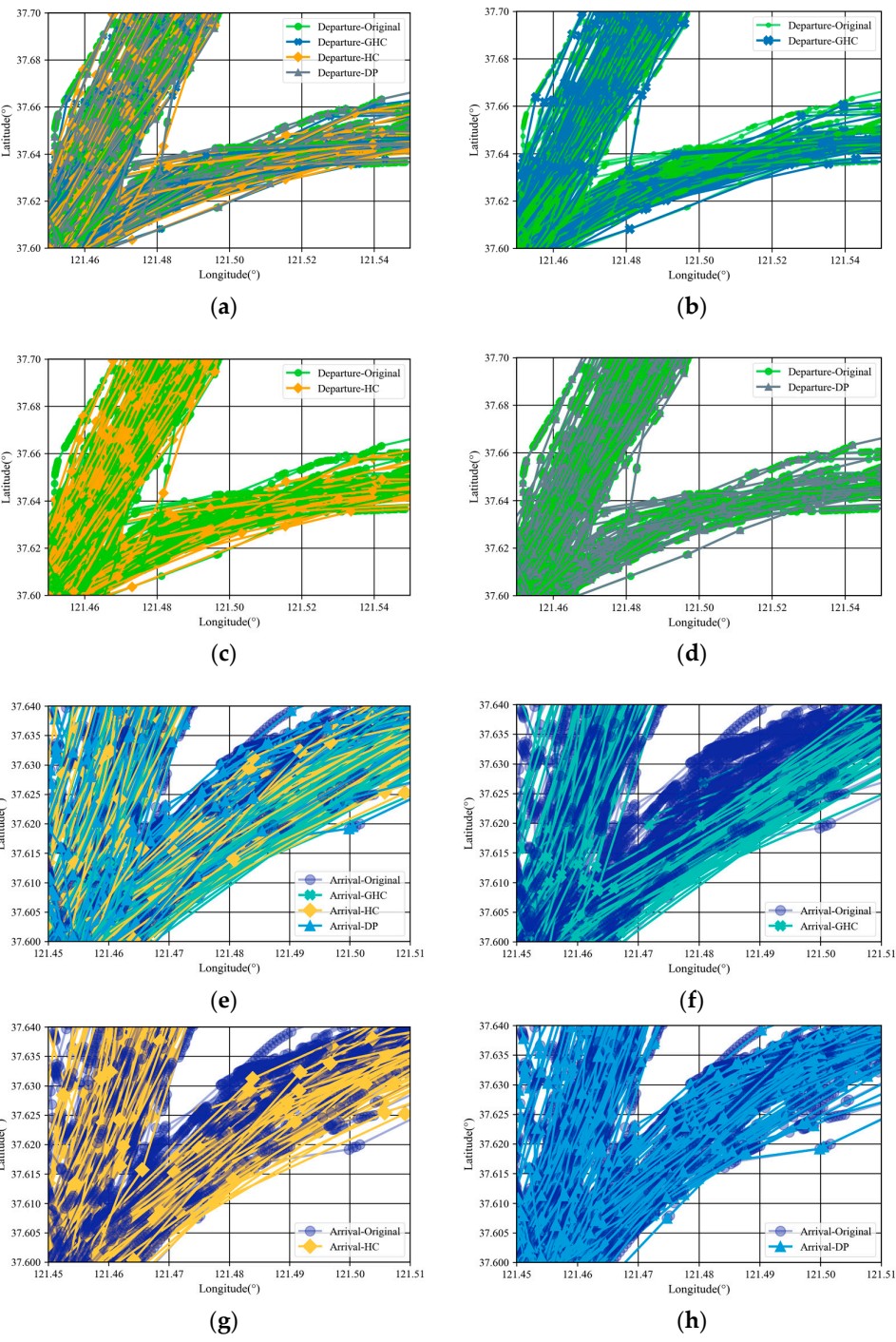

**Figure 12.** Trajectory map of the feature extraction: (**a**–**d**) show the comparison results of original trajectory, GHC feature, HC feature and DP feature of departure; (**e**–**h**) show the comparison results of original trajectory, GHC feature, HC feature and DP feature of arrival.

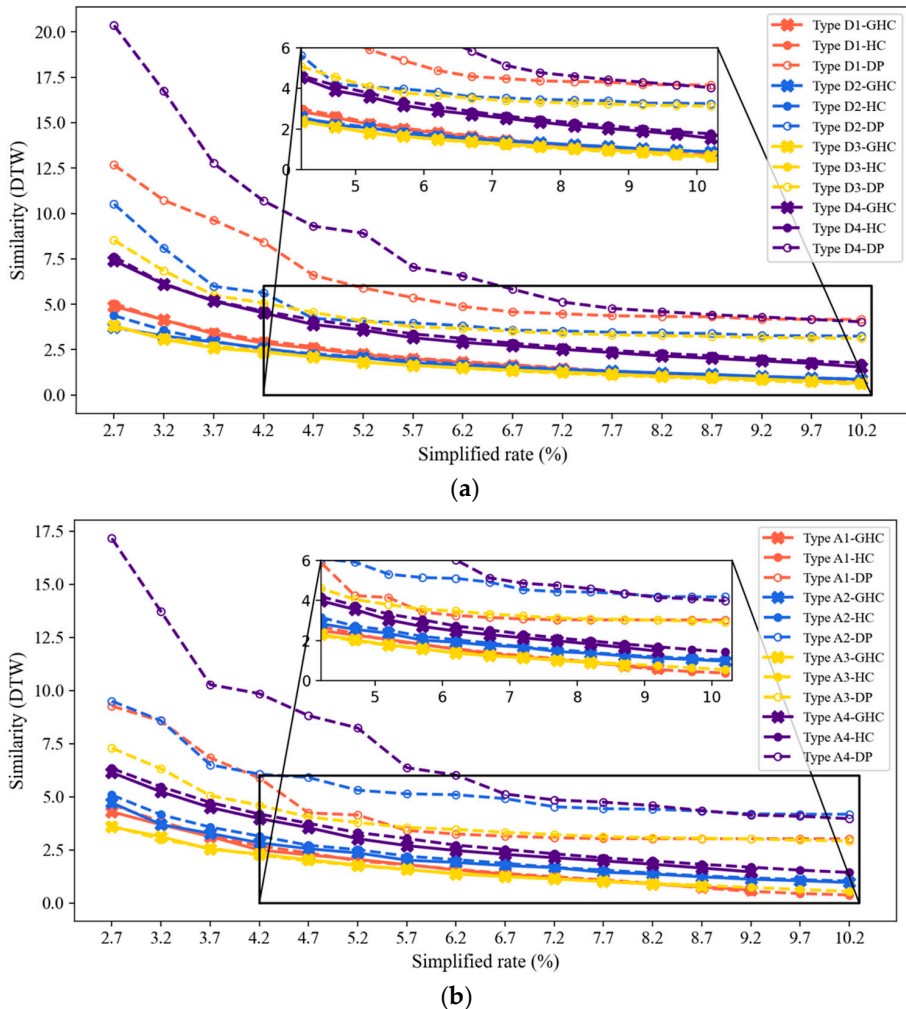

**Figure 13.** Similarity comparison diagram of the departure and arrival trajectory feature extraction: (**a**) similarity comparison of the departure trajectories; (**b**) similarity comparison of the arrival trajectories.

The experimental results show that the GHC algorithm proposed in this study was, in most cases, more similar to the original trajectory than the HC algorithm and DP algorithm in the feature extraction. From the specific experimental data, in the departure trajectory, the average similarity between the features extracted by GHC algorithm and the original trajectory was 2.31 when the trajectory points were guaranteed. Under the same conditions, the average similarity between the features extracted by the HC algorithm and the original trajectory was 2.37, and the average similarity between the features extracted by the DP algorithm and the original trajectory was 5.90. It can be seen from the results that the average similarity of the features extracted using the GHC algorithm was the best, which improved by 2.82% compared with the HC algorithm and 60.92% compared with the DP algorithm. In the arrival trajectory, the average similarity between the features extracted by the GHC algorithm and the original trajectory was 2.23 when the trajectory points were guaranteed. Under the same conditions, the average similarity between the features extracted by the HC algorithm and the original trajectory was 2.34, and the average similarity between the features extracted by the DP algorithm and the original trajectory was 5.47. It can be seen from the results that the average similarity of the feature extraction extracted using the GHC algorithm was the best, which improved by 4.68% compared with the HC algorithm and 59.26% compared with the DP algorithm. According to the data analysis, the GHC feature extraction method proposed in this study can better reduce the original data under the same simplification rate.

In order to optimize the optimal number of feature points extracted from the trajectory points, this study adopted the DTW similarity decline rate (i.e., loss decline rate) as the measurement method. Figure 14 shows the loss reduction rates of all trajectories in the departure and arrival trajectories under different simplification rates (i.e., the ratio of the number of feature extraction points to the average original trajectory points) of the three feature extraction algorithms. The average loss reduction rate data under different simplification rates are shown in Table 5. As can be seen from the figure, when the simplification rate of the GHC algorithm was 7.7%, the HC algorithm was 10.2%, the DP algorithm was 9.7%, and the loss reduction rate was the lowest. In the arrival trajectory, the reduction rate of the GHC algorithm was 9.7%, the reduction rate of the HC algorithm was 7.2%, and the loss reduction rate of the DP algorithm was the lowest when the reduction rate was 9.7%. Therefore, the above simplification rates were selected as the optimal number of feature extraction points. Thus, the corresponding reduction rate under the loss decline rate was selected as the optimal feature quantity.

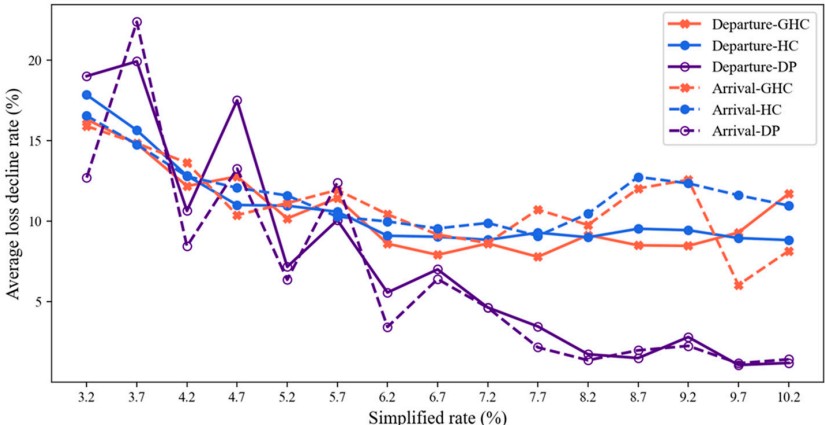

**Figure 14.** Average loss decline rate under different simplification rates.

**Table 5.** Average loss reduction rate data under different simplification rates.

| Simplified Rate | Departure | | | Arrival | | |
|---|---|---|---|---|---|---|
| | GHC | HC | DP | GHC | HC | DP |
| 3.2% | 16.31 | 17.83 | 19.00 | 15.90 | 16.54 | 12.70 |
| 3.7% | 14.80 | 15.64 | 19.91 | 14.83 | 14.76 | 22.36 |
| 4.2% | 12.18 | 12.82 | 10.64 | 13.62 | 12.78 | 8.45 |
| 4.7% | 12.77 | 11.01 | 17.49 | 10.39 | 12.07 | 13.26 |
| 5.2% | 10.17 | 10.96 | 7.17 | 11.12 | 11.59 | 6.37 |
| 5.7% | 11.44 | 10.59 | 10.07 | 11.94 | 10.28 | 12.40 |
| 6.2% | 8.61 | 9.10 | 5.58 | 10.43 | 9.99 | 3.44 |
| 6.7% | 7.91 | 9.04 | 7.01 | 9.17 | 9.55 | 6.41 |
| 7.2% | 8.62 | 8.85 | 4.65 | 8.65 | 9.89 | 4.64 |
| 7.7% | 7.79 | 9.29 | 3.48 | 10.72 | 9.09 | 2.18 |
| 8.2% | 9.14 | 9.00 | 1.75 | 9.76 | 10.47 | 1.38 |
| 8.7% | 8.51 | 9.53 | 1.51 | 12.03 | 12.75 | 1.99 |
| 9.2% | 8.47 | 9.44 | 2.82 | 12.57 | 12.35 | 2.27 |
| 9.7% | 9.30 | 8.95 | 1.09 | 6.05 | 11.61 | 1.19 |
| 10.2% | 11.70 | 8.83 | 1.21 | 8.15 | 10.97 | 1.44 |

### 4.5. Prediction of the Trajectory

In this section, the extracted trajectory features are used for the trajectory prediction. First, we divided the feature extraction data into a training set, test set, and verification set in the proportion of 70%, 20%, and 10%. The specific division method and sample quantity are shown in Table 6. The parameter selection of the model is very important, so we optimized the main parameters of the TRM model. In this study, we mainly adjusted the

number of heads of the multihead attention mechanism and time step. Among them, we set the selection range of the number of heads of the multihead attention mechanism to be 1–10, and the range of time step to be 1–10. The optimal result was taken as the parameters for the model. Figure 15 shows the tuning diagram of the TRM model's parameters. Through experiments, the model set the time step as 4 and the number of heads as 8. We set the initial learning rate as 0.01 and the initial epoch as 10,000. During the model training, the learning rate decrease of the algorithm was set. With the increase in the epoch, the learning rate changed to the initial 10% for every 100 epochs. In order to prevent overfitting, the early stop method was used, and the training was stopped when the training epoch was 100 and the loss was no longer decreasing.

**Table 6.** Data set division and sample number statistics.

| Data Set Partitioning | Ratio | Sample Size | |
| --- | --- | --- | --- |
| | | Departure | Arrival |
| Training set | 70% | 3580 | 4322 |
| Test set | 20% | 1023 | 1235 |
| Validation set | 10% | 512 | 618 |
| Total | 100% | 5115 | 6175 |

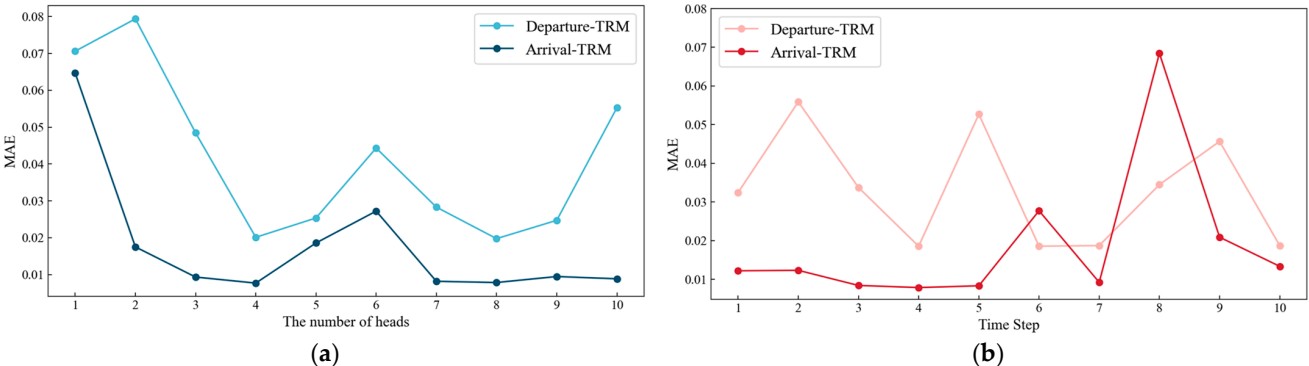

(**a**)  (**b**)

**Figure 15.** TRM model reference diagram: (**a**) number of heads; (**b**) time step.

The data of the departure and arrival trajectories extracted by the GHC algorithm were used as experimental data. Scatter plots of true and predicted values from the training set are shown in Figure 16. As can be seen the Figure 16, the model could well fit the longitude and latitude data in the trajectory. The evaluation indexes of the model in the test set are shown in Table 7. It can be seen from the table that the attention mechanism model had a better performance by the evaluation indicators.

**Table 7.** Evaluation indexes of test sets under different trajectory types.

| Direction of Movement | Type | Category | Indicators | | |
| --- | --- | --- | --- | --- | --- |
| | | | MAE | RMSE | $R^2$ |
| Departure | Type D1 | Longitude | 0.0272 | 0.0566 | 0.8686 |
| | | Latitude | 0.0050 | 0.0068 | 0.7570 |
| | Type D2 | Longitude | 0.0086 | 0.0165 | 0.8528 |
| | | Latitude | 0.0232 | 0.0453 | 0.8696 |
| | Type D3 | Longitude | 0.0281 | 0.0508 | 0.8657 |
| | | Latitude | 0.0187 | 0.0405 | 0.8638 |
| | Type D4 | Longitude | 0.0153 | 0.0266 | 0.7272 |
| | | Latitude | 0.0221 | 0.0515 | 0.8532 |

**Table 7.** *Cont.*

| Direction of Movement | Type | Category | Indicators | | |
| --- | --- | --- | --- | --- | --- |
| | | | **MAE** | **RMSE** | **R²** |
| Arrival | Type A1 | Longitude | 0.0112 | 0.0147 | 0.9910 |
| | | Latitude | 0.0038 | 0.0051 | 0.9211 |
| | Type A2 | Longitude | 0.0085 | 0.0117 | 0.9382 |
| | | Latitude | 0.0081 | 0.0097 | 0.9933 |
| | Type A3 | Longitude | 0.0040 | 0.0051 | 0.9828 |
| | | Latitude | 0.0084 | 0.0107 | 0.9920 |
| | Type A4 | Longitude | 0.0122 | 0.0151 | 0.9865 |
| | | Latitude | 0.0066 | 0.0082 | 0.9928 |

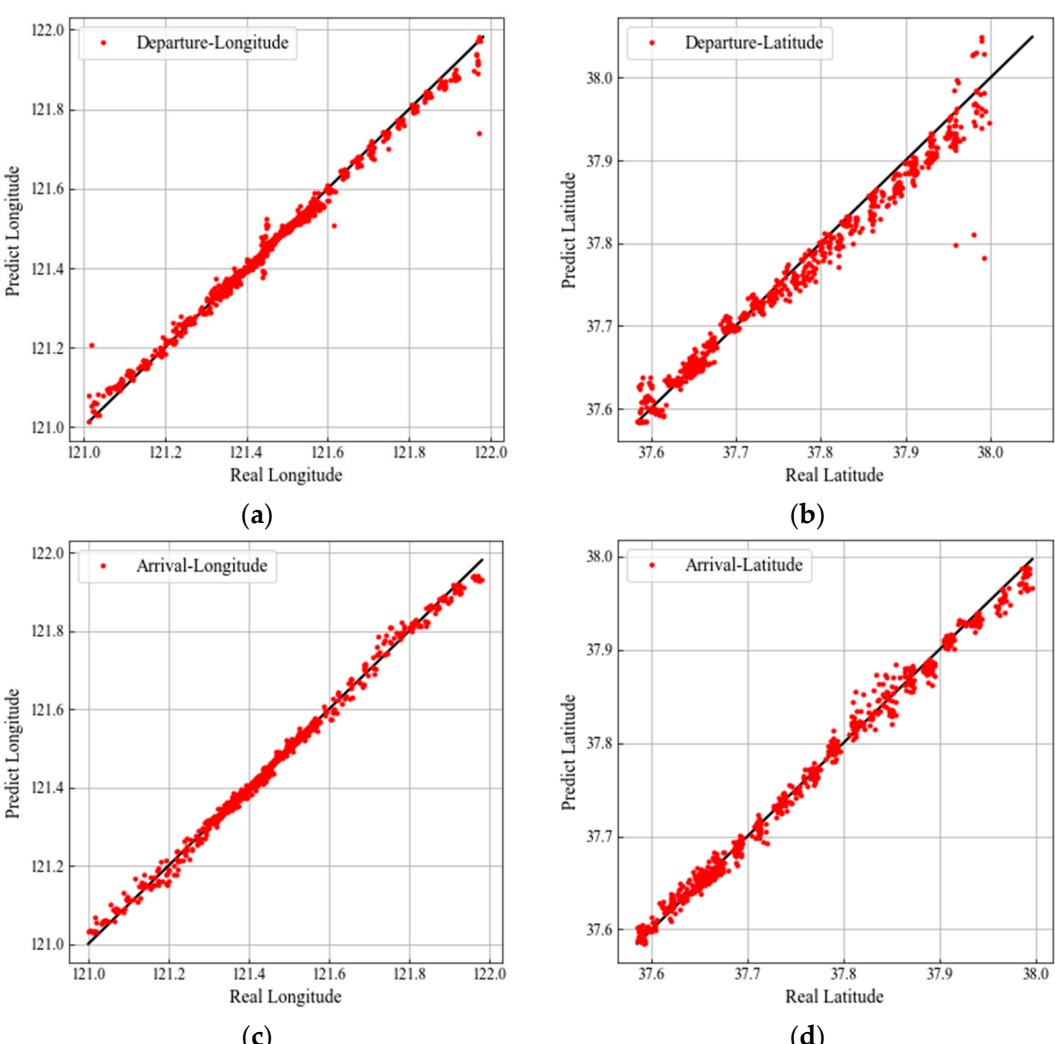

**Figure 16.** Scatter plots of real and predicted values from the training set: (**a**) departure longitude; (**b**) departure latitude; (**c**) arrival longitude; (**d**) arrival latitude.

4.5.1. Comparative Analysis of the Multi-Trajectory Prediction Results

In order to prove the superiority of the proposed prediction method, other traditional time series data prediction methods (such as LSTM and GRU) are used in this section for comparison. The selection of hyperparameters in the model will directly affect the experimental results. Therefore, it is necessary to optimize the hyperparameters of the model. Experimenting with all combinations of hyperparameters in the model results in a

longer experimental time. Therefore, only the major hyperparameters were considered. For the neural network model, we mainly considered the number of hidden layers, activation function, and time step. In terms of the number of hidden layers and the number of neurons, we considered five hidden layers at most and set the number of neurons from the first layer to the fifth layer as 100, 80, 40, 20, and 10, respectively. In terms of the activation function, we select relu, tanh, sigmoid, and linear as experimental objects. In terms of time step, we set the time step to be between 1 and 10 for the optimization. The combination comparison of the number of different hidden layers, the number of neurons, and different activation functions is shown in Figure 17. Figure 18 shows the result of time step parameter adjustment. The optimal parameters obtained through the experiment are shown in Table 8. The experimental results are compared and analyzed. Figure 19 shows the performance of the MAE of the different models on the different trajectories. The MAE of the different prediction methods on different types of trajectories is shown in Table 9.

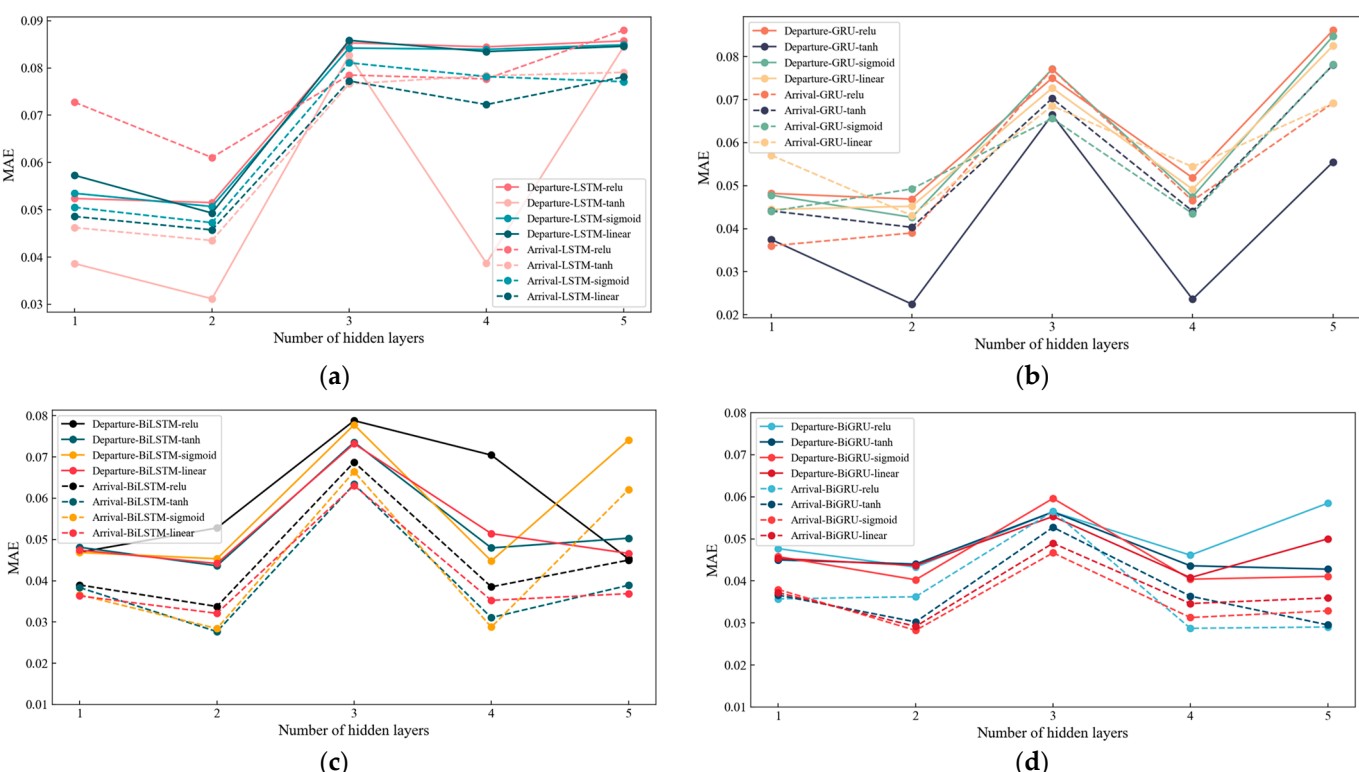

**Figure 17.** The combination comparison of the number of different hidden layers, the number of neurons, and different activation functions: (**a**) LSTM; (**b**) GRU; (**c**) BiLSTM; (**d**) BiGRU.

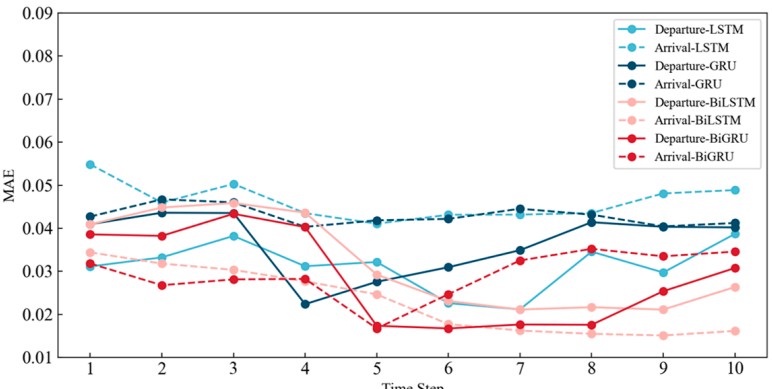

**Figure 18.** Time step parameter adjustment diagram.

**Table 8.** Optimal parameters for the different models.

| Model | Hyperparameter Setting |
|---|---|
| LSTM | Number of Hidden Layers = 2<br>Number of Neurons = 10,080<br>Activation Function = tanh<br>Batch Size = 4<br>Departure Time Steps = 7, Arrival Time Steps = 5<br>Dropout = 0.1 |
| GRU | Number of Hidden Layers = 2<br>Number of Neurons = 10,080<br>Activation Function = tanh<br>Batch Size = 4<br>Departure Time Steps = 4, Arrival Time Steps = 4<br>Dropout = 0.1 |
| BiLSTM | Number of Hidden Layers = 2<br>Number of Neurons = 10,080<br>Activation Function = tanh<br>Batch Size = 4<br>Departure Time Steps = 9, Arrival Time Steps = 9<br>Dropout = 0.1 |
| BiGRU | Number of Hidden Layers = 2<br>Number of Neurons = 10,080<br>Activation Function = sigmoid<br>Batch Size = 4<br>Departure Time Steps = 6, Arrival Time Steps = 5<br>Dropout = 0.1 |

**Table 9.** MAEs of the results predicted by the different methods.

| Direction of Movement | Type | Category | Method | | | | |
|---|---|---|---|---|---|---|---|
| | | | TRM | LSTM | GRU | BiLSTM | BiGRU |
| Departure | Type D1 | Longitude<br>Latitude | 0.0272<br>0.0050 | 0.0321<br>0.0067 | 0.0363<br>0.0068 | 0.0305<br>0.0072 | 0.0337<br>0.0063 |
| | Type D2 | Longitude<br>Latitude | 0.0086<br>0.0232 | 0.0082<br>0.0282 | 0.0100<br>0.0289 | 0.0094<br>0.0264 | 0.0091<br>0.0273 |
| | Type D3 | Longitude<br>Latitude | 0.0281<br>0.0187 | 0.0309<br>0.0229 | 0.0318<br>0.0243 | 0.0295<br>0.0241 | 0.0328<br>0.0271 |
| | Type D4 | Longitude<br>Latitude | 0.0153<br>0.0221 | 0.0155<br>0.0246 | 0.0156<br>0.0258 | 0.0156<br>0.0255 | 0.0158<br>0.0284 |
| Arrival | Type A1 | Longitude<br>Latitude | 0.0112<br>0.0038 | 0.0308<br>0.0074 | 0.0329<br>0.0072 | 0.0254<br>0.0070 | 0.0287<br>0.0066 |
| | Type A2 | Longitude<br>Latitude | 0.0085<br>0.0081 | 0.0126<br>0.0249 | 0.0108<br>0.0241 | 0.0110<br>0.0188 | 0.0117<br>0.0211 |
| | Type A3 | Longitude<br>Latitude | 0.0040<br>0.0084 | 0.0075<br>0.0222 | 0.0078<br>0.0228 | 0.0071<br>0.0190 | 0.0072<br>0.0199 |
| | Type A4 | Longitude<br>Latitude | 0.0122<br>0.0066 | 0.0250<br>0.1990 | 0.0250<br>0.1920 | 0.0153<br>0.0176 | 0.0206<br>0.0179 |

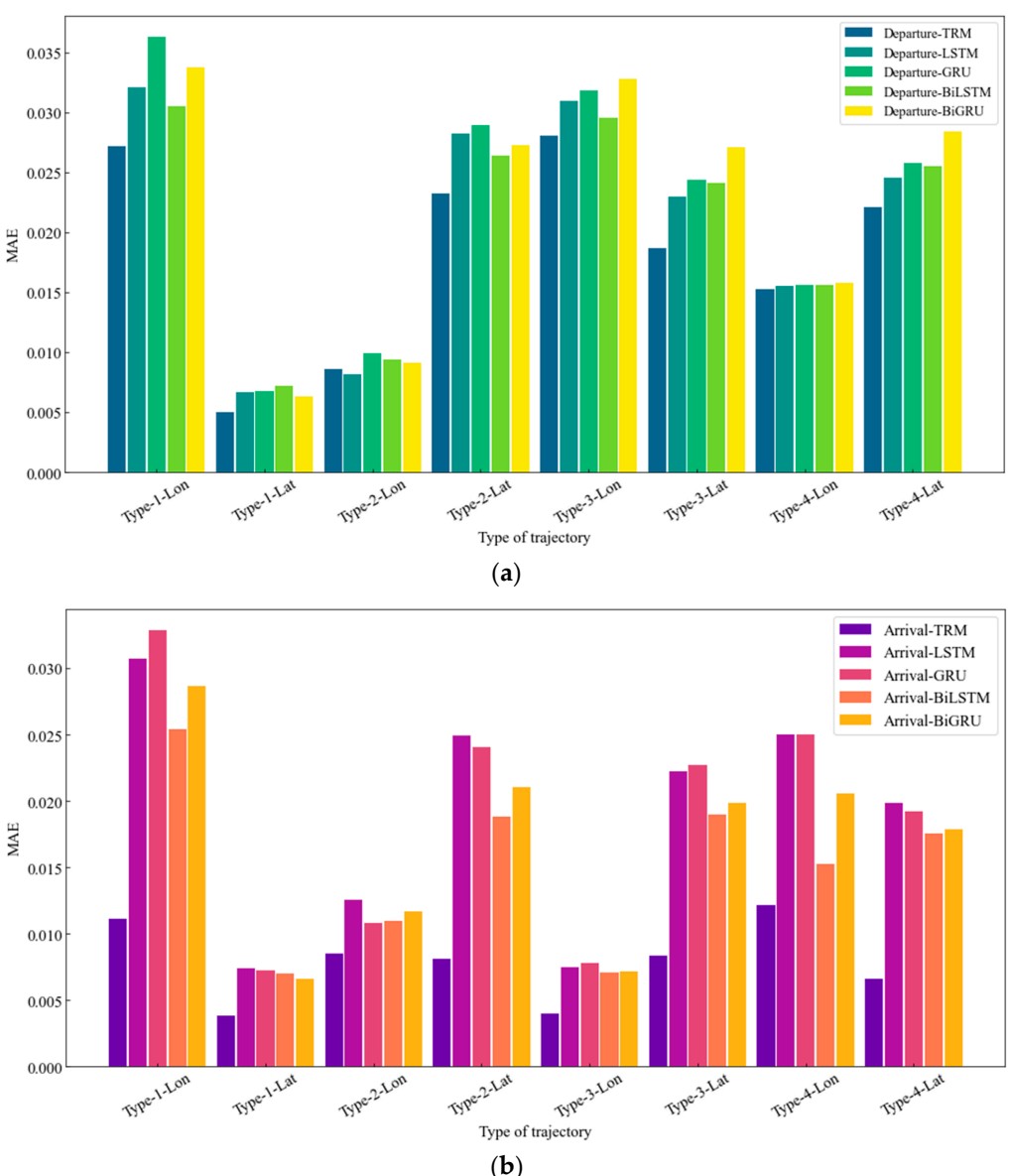

**Figure 19.** MAE histogram of the prediction results by the different methods: (**a**) comparison of the departure trajectory models; (**b**) comparison of the arrival trajectory models.

As can be seen from Figure 19, the TRM model was generally superior to the LSTM and GRU models in the performance of the MAE. From the concrete data results, firstly, the prediction results of the TRM model were compared with the LSTM model. The TRM longitude evaluation index was slightly higher than that of the LSTM in departure trajectory type 2. In other cases, the prediction evaluation index of the TRM model was all due to the evaluation index of the LSTM. In the departure trajectory, the MAE of the TRM was 8.65% lower than that of the LSTM in terms of the longitude and 16.26% lower than that of the LSTM in terms of the latitude. In the arrival trajectory, the MAE of the TRM was 52.70% lower than that of the LSTM in terms of the longitude and 63.84% lower than that of the LSTM in terms of the latitude. Secondly, the TRM model was compared with the GRU model. The TRM model was superior to the GRU model in the MAE results. In the departure trajectory, the MAE of the TRM model was 15.48% lower than that of the GRU in terms of the longitude and 19.58% lower than that of the GRU in terms of the latitude. In the arrival trajectory, the MAE of the TRM was 53.07% lower than that of the GRU for the longitude and 62.71% lower than that of the GRU for the latitude. Thirdly, the TRM model was compared with the BiLSTM model. In the departure trajectory, the MAE of the

TRM model was 6.82% lower than that of the BiLSTM in terms of the longitude and 17.07% lower than that of the BiLSTM in terms of the latitude. In the arrival trajectory, the MAE of the TRM was 38.95% lower than that of the BiLSTM for the longitude and 56.89% lower than that of the BiLSTM for the latitude. Finally, we compared the BiGRU model with the TRM model. In the departure trajectory, the MAE of the TRM model was 13.35% lower than that of the BiGRU in terms of the longitude and 22.56% lower than that of the BiGRU in terms of the latitude. In the arrival trajectory, the MAE of the TRM was 47.36% lower than that of the BiGRU for the longitude and 58.93% lower than that of the BiGRU for the latitude. Therefore, it can be seen from the specific data that the TRM model had a better predictive performance.

4.5.2. Comparison the Different Models on New Trajectories

In order to better verify the prediction effect of the model on a new trajectory, the feature data extracted by the GHC algorithm was used as the training set, the new trajectory was selected as the test set for the test, and the LSTM and GRU models were used for the comparison test. The MAE evaluation indexes of the different models for the multi-trajectory prediction are shown in Table 10. As can be seen from the Table 10, the TRM model was superior to the LSTM and GRU models in terms of the prediction and evaluation indicators. According to the data analysis, in the departure trajectory, the MAE of the TRM model was 5.51% lower in the longitude than the LSTM, 13.24% lower than the GRU, 26.55% lower than the BiLSTM, 11.51% lower than the BiGRU, 0.54% lower in the latitude than LSTM, and 7.36% lower in the GRU, 4.96% lower than the BiLSTM, and 12.84% lower than the BiGRU. In the arrival trajectory, the MAE of the TRM model was 3.82% lower in the longitude than that of the LSTM and 5.10% lower than that of the GRU, 4.41% lower than the BiLSTM, and 2.65% lower than the BiGRU. In the latitude, it was 6.83% lower than the LSTM and 16.80% lower than the GRU, 6.01% lower than the BiLSTM, and 2.36% lower than the BiGRU. It can be seen from the results that the TRM model had a certain generalization performance in the new trajectory prediction, which proves that the TRM model can achieve the expected function in ship multi-trajectory prediction.

**Table 10.** MAEs of the different prediction methods applied to new trajectories.

| Model | Category | Type | |
| --- | --- | --- | --- |
| | | Departure | Arrival |
| LSTM | Longitude | 0.0554 | 0.0992 |
| | Latitude | 0.0695 | 0.0889 |
| GRU | Longitude | 0.0603 | 0.1005 |
| | Latitude | 0.0747 | 0.0996 |
| BiLSTM | Longitude | 0.0712 | 0.0998 |
| | Latitude | 0.0728 | 0.0882 |
| BiGRU | Longitude | 0.0591 | 0.0980 |
| | Latitude | 0.0701 | 0.0849 |
| TRM | Longitude | 0.0523 | 0.0954 |
| | Latitude | 0.0692 | 0.0829 |

4.5.3. Comparison of the Different Feature the Extraction Methods

In order to verify the effectiveness of the GHC feature extraction method for model training, this study used the GHC, HC, and DP algorithms for the feature extraction of the same set of data. The data after feature extraction were selected to train different models, and new trajectories were selected for testing. The MAE evaluation indexes of the combination of different feature extraction methods and models are shown in Table 11, and the histogram of the evaluation indexes of the different models is shown in Figure 20. It can be seen from Figure 20 and Table 11 that the combination of features extracted using the GHC algorithm and different models was the best for the MAE compared with other combinations, which indicates that the GHC feature extraction algorithm effectively

improves the training effect of the model. It is worth noting that the GHC-TRM combined model achieves the best measure in all the algorithms. This proves the advantages of the GHC-TRM model in the trajectory prediction effect and also proves the feasibility of the multi-trajectory prediction framework proposed in this study for trajectory prediction.

**Table 11.** MAE comparison table of the different combinations of methods.

| Model | Category | Type | |
| --- | --- | --- | --- |
| | | Departure | Arrival |
| DP-LSTM | Longitude | 0.0570 | 0.1012 |
| | Latitude | 0.0795 | 0.0961 |
| HC-LSTM | Longitude | 0.0583 | 0.1336 |
| | Latitude | 0.0758 | 0.1225 |
| GHC-LSTM | Longitude | 0.0554 | 0.0992 |
| | Latitude | 0.0695 | 0.0889 |
| DP-GRU | Longitude | 0.0619 | 0.1018 |
| | Latitude | 0.0765 | 0.1085 |
| HC-GRU | Longitude | 0.0662 | 0.1424 |
| | Latitude | 0.0753 | 0.1404 |
| GHC-GRU | Longitude | 0.0603 | 0.1005 |
| | Latitude | 0.0747 | 0.0996 |
| DP-BiLSTM | Longitude | 0.0836 | 0.1014 |
| | Latitude | 0.0788 | 0.0933 |
| HC-BiLSTM | Longitude | 0.0994 | 0.1068 |
| | Latitude | 0.0850 | 0.0991 |
| GHC-BiLSTM | Longitude | 0.0712 | 0.0998 |
| | Latitude | 0.0728 | 0.0882 |
| DP-BiGRU | Longitude | 0.0785 | 0.1204 |
| | Latitude | 0.0913 | 0.0854 |
| HC-BiGRU | Longitude | 0.1016 | 0.0985 |
| | Latitude | 0.0913 | 0.1090 |
| GHC-BiGRU | Longitude | 0.0591 | 0.0980 |
| | Latitude | 0.0701 | 0.0849 |
| DP-TRM | Longitude | 0.0782 | 0.1079 |
| | Latitude | 0.0957 | 0.0917 |
| HC-TRM | Longitude | 0.0562 | 0.1140 |
| | Latitude | 0.0990 | 0.1066 |
| GHC-TRM | Longitude | 0.0523 | 0.0954 |
| | Latitude | 0.0692 | 0.0829 |

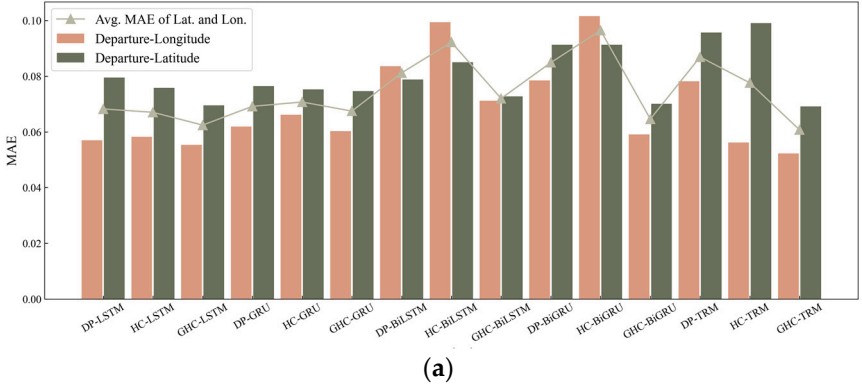

**(a)**

**Figure 20.** *Cont*.

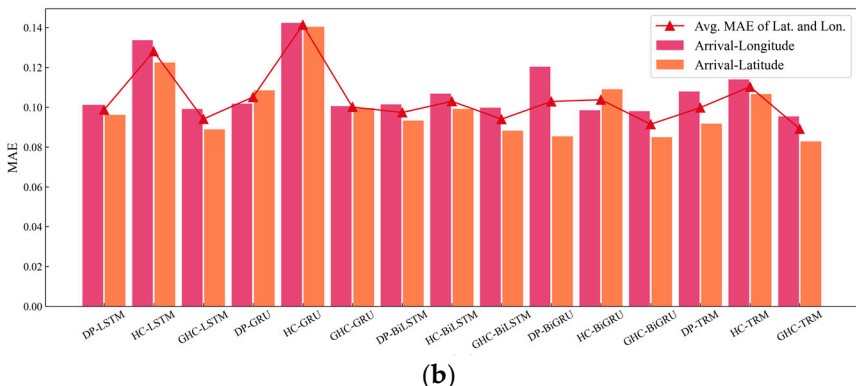

**(b)**

**Figure 20.** MAE histogram of the different methods combined: (**a**) comparison of the departure trajectory models; (**b**) comparison of the arrival trajectory models.

## 5. Conclusions

Compared with several existing methods, in order to make up for the AIS data collection system's dense defects at a certain point caused by the uneven collection times during data collection, solve the traditional feature extraction methods' high sensitivity at the trajectory dense point, and make up for the defects in the prediction accuracy of the multi-trajectory prediction model, in this study, we proposed a multi-trajectory hierarchical clustering method and a multi-trajectory prediction framework based on an attention mechanism. The key idea was found through AIS data visualization that the inbound and outbound trajectories have obvious grouping characteristics. On this basis, it was proposed that intra-group features should be extracted and the motion mode of similar trajectories at the same position when the trajectory points merged should be considered. Therefore, the features of each trajectory are not only affected by its own trajectory points, but also by the trajectory points of similar trajectories. To a certain extent, the sensitivity of the trajectory density points was eliminated. The input weight was calculated through the attention mechanism model to improve the accuracy of the multi-trajectory prediction. The comprehensive evaluation of the case study shows that the proposed feature extraction method was more similar to the original trajectory, the trajectory shape was better restored, and the proposed prediction model achieved satisfactory prediction accuracy. However, the proposed feature extraction method and prediction model still have defects in similarity and accuracy in the regions with more turns. In future research work, the feature extraction and prediction of multi-steering regions need further study.

**Author Contributions:** Writing—review, editing, methodology, and formal analysis, Y.Z.; data curation and writing—original draft preparation, J.J.; validation, J.J. All authors have read and agreed to the published version of the manuscript.

**Funding:** This work was supported in part by the National Natural Science Foundation of China (grant nos. 52131101 and 51939001), the LiaoNing Revitalization Talents Program (grant no. XLYC1807046), and the Science and Technology Fund for Distinguished Young Scholars of Dalian (grant no. 2021RJ08).

**Institutional Review Board Statement:** Not applicable.

**Informed Consent Statement:** Not applicable.

**Data Availability Statement:** Data can be obtained via request to the corresponding author.

**Conflicts of Interest:** The authors declare that there are no conflicts of interest regarding the publication of this paper.

## Appendix A

The DP algorithm is shown in Figure A1, and the specific steps are as follows:

(1) Connect the trajectory to be processed with a straight line from end to end. Set this line as the initial simplified line;

(2) Find the maximum distance metric from the simplified line $d_{max}$;

(3) Compare $d_{max}$ with the thinning threshold. If $d_{max} < threshold$, delete all the intermediate points on this curve. If $d_{max} \geq threshold$, divide the curve into two parts;

(4) Repeat the above steps until the distance metric from all points to the reduced line is less than threshold.

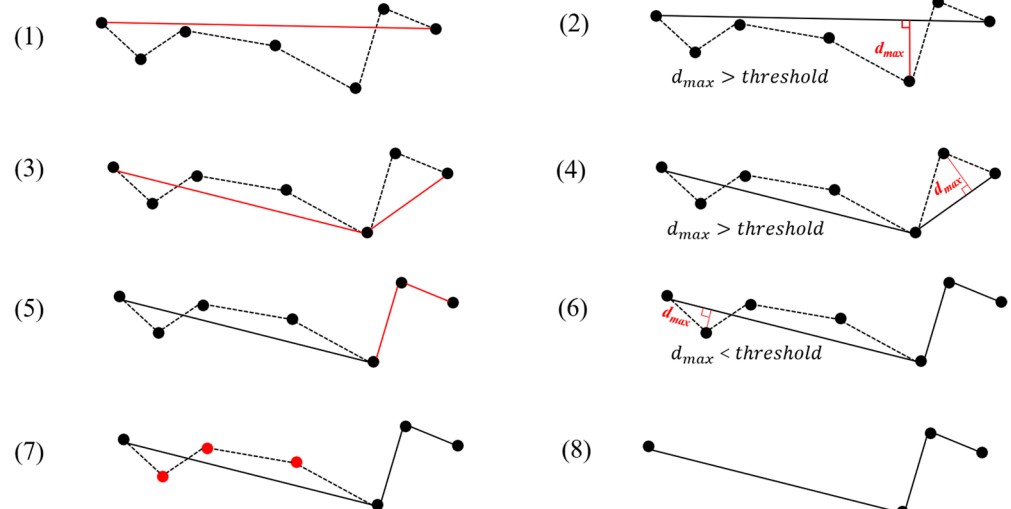

**Figure A1.** Paradigm of the DP algorithm.

The HC algorithm is shown in Figure A2, and the processing steps are as follows:

(1) Calculate the Euclidean metric $d_i$ between two adjacent points. Save the $d_i$ to the similarity set *S*;

(2) Find two points of the minimum Euclidean metric;

(3) Take the average of two points of the minimum Euclidean metric in the set *S* to replace the original point in the trajectory set;

(4) Repeat steps (1)–(3) until the number of trajectory points converges to the target point *k*.

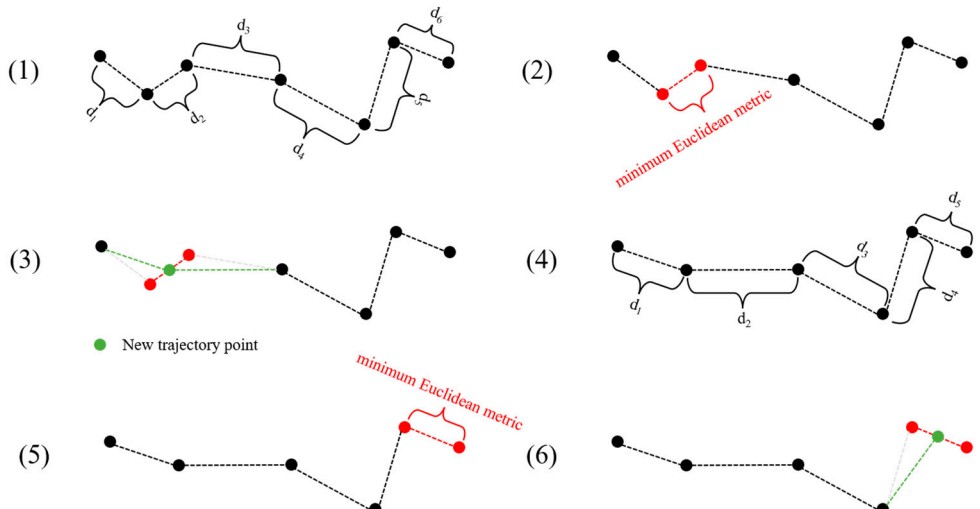

**Figure A2.** Paradigm of the agglomerative hierarchical clustering algorithm.

The RNN structure is shown in Figure A3. The working principle of an RNN is as follows:

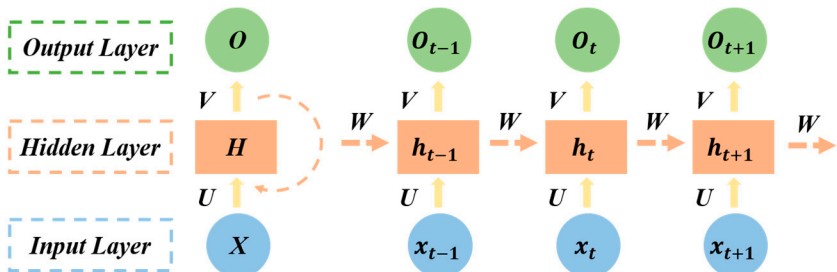

**Figure A3.** Structure of the recurrent neural network.

The principles from the input layer to the hidden layer and then to the output layer are shown in Equations (A1) and (A2), respectively.

$$h_t = f_1(Ux_t + Wh_{t-1}) \tag{A1}$$

$$O_t = f_2(Vh_t) \tag{A2}$$

where $h_t$ represents the output of neurons in the hidden layer at time $t$; $U$, $V$, and $W$ are the connection weight matrices that connect the relationships between input layers, hidden layers, and output layers; $O_t$ represents the output; and $f_1(x)$ and $f_2(x)$ are the activation functions.

The LSTM network structure is shown in Figure A4.

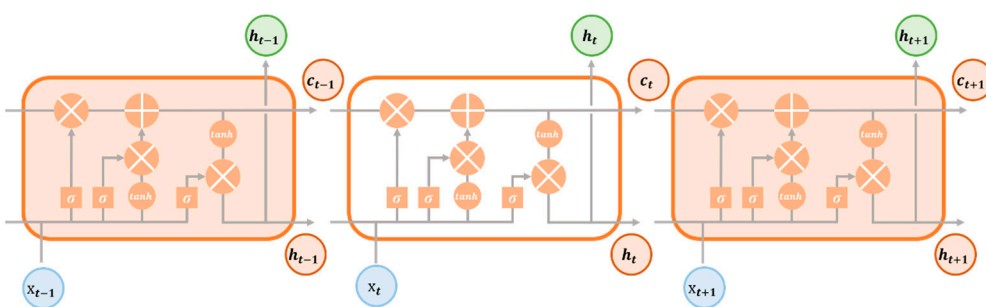

**Figure A4.** Structure of the long short-term memory neural network.

The specific working principle of each part of the LSTM is as follows:

The forgetting gate determines the degree of retention of the trajectory information.

$$f_t = \sigma\left(W_f \times [h_{t-1}, X_t] + b_f\right) \tag{A3}$$

where $W_f$ is the weight matrix of the forgetting gate; $b_f$ is the bias of the forgetting gate; and $\sigma$ is the activation function.

The input gate determines the information that needs to be updated.

$$i_t = \sigma(W_i \times [h_{t-1}, X_t] + b_i) \tag{A4}$$

$$\widetilde{C}_t = \tanh(W_c \times [h_{t-1}, X_t] + b_c) \tag{A5}$$

$$C_t = f_t \times C_{t-1} + i_t \times \widetilde{C}_t \tag{A6}$$

where notation $W_i$ is the weight matrix of the input gate; $W_c$ is the weight matrix for calculating the cell state; $b_i$ and $b_c$ are the bias; and tan h is the hyperbolic tangent activation function.

The output gate obtains the output of the unit layer.

$$O_t = \sigma(W_o \times [h_{t-1}, X_t] + b_o) \tag{A7}$$

$$h_t = O_t \times \tanh(c_t) \tag{A8}$$

where $O_t$ is the output of the output gate at time $t$, and ht is the output of the local LSTM unit.

A diagram of the structure of the GRU is shown in Figure A5.

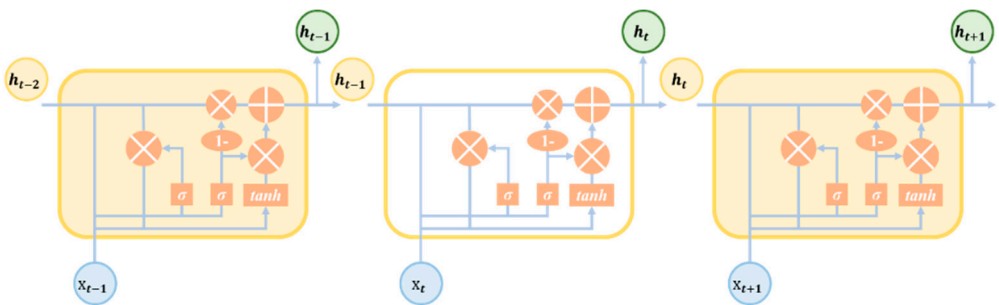

**Figure A5.** Structure of the gate recurrent unit.

The update gate $z_t$ is used to divide the proportion of state information at time $t - 1$ to the current time $t$. The reset gate state $r_t$ and the update gate state $z_t$ are calculated by Equations (A9) and (A10):

$$r_t = \sigma(x_t w_r + h_{t-1} W_r + b_r) \tag{A9}$$

$$z_t = \sigma(x_t w_z + h_{t-1} W_z + b_z) \tag{A10}$$

where $w_r$, $W_r$, $w_z$, and $W_z$ are the weight coefficient matrices of $r_t$ and $z_t$. The notations $b_r$ and $b_z$ are the bias quantities of $r_t$ and $z_t$, respectively. The notation $h_{t-1}$ is the state at time $t - 1$.

After obtaining the $r_t$ and $z_t$ states, the temporary state $\widetilde{h}_t^i$, from $h_{t-1}$ through the state at time $t - 1$ is calculated, and the current time $t$ output state is obtained.

$$\widetilde{h}_t^i = tanh(w_{x_t} + U(r_t \cdot h_{t-1}))^i \tag{A11}$$

$$h' = tanh(x_t w_h + W_h(h_{t-1} \cdot r_t) + b_h) \tag{A12}$$

$$h_t = (1 - z_t) \cdot h_{t-1} + z_t \cdot h' \tag{A13}$$

The notations $w_h$ and $W_h$ are the weights of the temporary state $h'$. The notation $b_h$ is the bias of the temporary state $h'$.

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
