# Peer review of "Prediction of Ship Trajectory in Nearby Port Waters Based on Attention Mechanism Model"

_sustainability, doi:10.3390/su15097435_

Round 1

Reviewer 1 Report

Please see my attached

Author Response

Thank you very much for your time and effort on our manuscript. We have completely revised the manuscript according to the comments, and listed the response one by one in the attachment. Please see the attached file.

Reviewer 2 Report

The article deals with the interesting and topical issue of determining and predicting ship trajectories. The overall structure of the article is correct and not objectionable. The article contains a satisfactory analysis of solutions presented in the literature on the subject and properly defines the research problem being solved. However, the shortcomings present in it, both in terms of content and editing, make it inadequate for publication in its current state.

Substantive comments:

Calculating distances using equation (2), taking longitude and latitude as x and y respectively, is an error and makes the results obtained and presented in the paper unreliable.

In many places in the article, units are missing (e.g. lines 23, 388, 437), making it difficult or impossible to interpret the results.

It is true that AIS and ECDIS information is insufficient to navigate (lines 32-34), so decisions are made by officers (generally not pilots) based on visual observation and, in restricted visibility, radar observation. In addition, the way ships behave is regulated by regulations (COLREG). The prediction of the trajectory of ships, under the current regulations, can only be an auxiliary aid to navigation.

Editorial remarks:

·         Some of the drawings (e.g. latitude and longitude distributions fig. 14) are meaningless and make the other drawings by their small size hardly visible.

·         Some abbreviations are used without giving the full name or the full name is only given later in the article.

·         Lack of consistency in the use of notation of names and terms (e.g. D-P and DP, heading and course).

·         Overall the article needs further editing work as there are many mistakes and omissions:

E.g.:

·         Line 43 - author's surname missing

·         No reference in the text to items 6,7,8 from the list of references

·         Notation of numbers not in line with the generally accepted standard (e.g. line 99)

·         The description of figure 4 contains terms that do not appear in the figure (e.g. tanh)

·         Line 85-87: same sentence twice (first with error)

·         Line 247 - unnecessary division into two sentences

·         .. and so on

Author Response

(The authors gave the same response as above.)

Reviewer 3 Report

Authors have used old data and known methods without proper state-of-the-art comparison, proving the novelty of the work insignificant. Please find my comments for improvisation:  

1.       The manuscript is full of much-known text. The authors have included a detailed explanation of earlier works but that is not required. It is not a review-based work.

2.       Authors have given a reference [17][18] and illustrated before their methodology. As the reference works are also related to ships and regression-based work, then it should be clarified.

3.       In Table 6, different hyper-parameters are given. Do the authors have taken any steps for hyper-parameter optimization or are these randomly selected?

4.       The quality of many graphs is poor. It should be enhanced.

5.       The authors have analyzed the data from January to June 2019, why not the recent one? It would be beneficial to study the current trend, not the old one. Also, the data count is less.

6.       DBSCAN is well known, and it is good but authors should explore other clustering approaches to compare the performance.

7.       Authors should give some strong reasons why authors did not use the recent regression-based approaches if they are performing well on problems of similar context.

8.       A good state-of-the-art comparison is missing.

9.       How is data split into different sets, i.e., Train and Test Set? Is cross-validation adapted?

Author Response

(The authors gave the same response as above.)

Round 2

Reviewer 1 Report

Thank you for your response, this reviewer has no further comments

Author Response

We really appreciate the reviewers for the valuable comments and helpful suggestions on our manuscript.

Reviewer 2 Report

Most of the comments have been taken into account, thus improving the quality of the article. However, I still cannot agree with the use of equation (2) to calculate the distance between points. The explanation that it was used "to measure the similarity between each pair of AIS points" is not sufficient. The distance between two points lying on the same meridian or latitude line, with the same d(lon) or d(lat) given in degrees, minutes, etc. is not the same, so it is difficult to use these results to study the similarity of points. In other words, d(lon) cannot be treated as a distance because it depends on the latitude. Additionally, the article still, mentions the distance between points (line 162).

Author Response

We really appreciate the reviewer for the valuable comments and helpful suggestions on our manuscript. The manuscript has been completely revised according to the comments and suggestions of Reviewer 2. The detailed responses and revisions were listed in the attachment.

Reviewer 3 Report

All my comments are well addressed. The manuscript can be accepted for publication.

Author Response

(The authors gave the same response as above.)

Round 3

Reviewer 2 Report

I still think that it would be more correct to use the orthodromic (Great Circle) distance formulas, or at least multiply the difference in the longitude of the points times the cosine of the mean latitude, but I guess that would require recalculation of all results. I believe that the obtained results would not differ much (but they would be more substantively correct), so in my opinion, the paper can be published in the present form.